# Glutamine synthetase mRNA releases sRNA from its 3′UTR to regulate carbon/nitrogen metabolic balance in *Enterobacteriaceae*

**Masatoshi Miyakoshi[1,2,3]\*, Teppei Morita[4,5], Asaki Kobayashi[2], Anna Berger[3], Hiroki Takahashi[6], Yasuhiro Gotoh[7], Tetsuya Hayashi[7], Kan Tanaka[8]**

[1]Department of Infection Biology, Faculty of Medicine, University of Tsukuba, Tsukuba, Japan; [2]Transborder Medical Research Center, University of Tsukuba, Tsukuba, Japan; [3]International Joint Degree Master's Program in Agro-Biomedical Science in Food and Health (GIP-TRIAD), University of Tsukuba, Tsukuba, Japan; [4]Institute for Advanced Biosciences, Keio University, Tsuruoka, Japan; [5]Graduate School of Media and Governance, Keio University, Fujisawa, Japan; [6]Medical Mycology Research Center, Chiba University, Chiba, Japan; [7]Department of Bacteriology, Faculty of Medical Sciences, Kyushu University, Fukuoka, Japan; [8]Laboratory for Chemistry and Life Science, Institute of Innovative Research, Tokyo Institute of Technology, Yokohama, Japan

**\*For correspondence:**
mmiyakoshi@md.tsukuba.ac.jp

**Competing interest:** The authors declare that no competing interests exist.

**Abstract** Glutamine synthetase (GS) is the key enzyme of nitrogen assimilation induced under nitrogen limiting conditions. The carbon skeleton of glutamate and glutamine, 2-oxoglutarate, is supplied from the TCA cycle, but how this metabolic flow is controlled in response to nitrogen availability remains unknown. We show that the expression of the E1o component of 2-oxoglutarate dehydrogenase, SucA, is repressed under nitrogen limitation in *Salmonella enterica* and *Escherichia coli*. The repression is exerted at the post-transcriptional level by an Hfq-dependent sRNA GlnZ generated from the 3′UTR of the GS-encoding *glnA* mRNA. Enterobacterial GlnZ variants contain a conserved seed sequence and primarily regulate *sucA* through base-pairing far upstream of the translation initiation region. During growth on glutamine as the nitrogen source, the *glnA* 3′UTR deletion mutants expressed SucA at higher levels than the *S. enterica* and *E. coli* wild-type strains, respectively. In *E. coli*, the transcriptional regulator Nac also participates in the repression of *sucA*. Lastly, this study clarifies that the release of GlnZ from the *glnA* mRNA by RNase E is essential for the post-transcriptional regulation of *sucA*. Thus, the mRNA coordinates the two independent functions to balance the supply and demand of the fundamental metabolites.

## Editor's evaluation

This important study supports the role of a regulatory RNA to adjust to the changing availability of nitrogen by controlling expression of key enzymes of the nitrogen assimilation pathway in *Enterobacteriaceae*. The evidence supporting the conclusions is convincing, with state-of-the-art genetic and biochemical assays. The work will be of interest to scientists within the field of microbial gene regulation.

## Introduction

Nitrogen is an essential element of the cell. Ammonia is the energetically most preferable nitrogen source and is assimilated via either glutamate dehydrogenase (GDH) or the glutamine synthetase (GS) and glutamine 2-oxoglutarate amidotransferase (GOGAT) pathway (*Reitzer, 2003*). GDH catalyzes the reductive amination of 2-oxoglutarate (2-OG, a.k.a. α-ketoglutarate) to glutamate (Glu) without ATP consumption. In contrast, GS catalyzes the amidation of Glu to form glutamine (Gln) using one molecule of ATP, and then GOGAT transfers the amide group reductively to 2-OG to generate two Glu molecules, yielding one net Glu from 2-OG (*Figure 1A*).

Nitrogen is limiting for most bacteria in freshwater, marine, and terrestrial ecosystems (*Elser et al., 2007*) and mammalian large intestines (*Reese et al., 2018*). To deal with the environmental availability of nitrogen, enterobacteria such as *Salmonella enterica* and *Escherichia coli* have developed a complex regulatory network (*van Heeswijk et al., 2013*). This regulation is particularly crucial for the facultative intracellular pathogen *S. enterica* since the GS-encoding *glnA* gene is essential during invasion and proliferation in the host cells (*Popp et al., 2015*; *Klose and Mekalanos, 1997*; *Aurass et al., 2018*). The *glnA* gene constitutes an operon along with the *glnLG* genes encoding the NtrB/NtrC (GlnL/GlnG) two-component regulator system, which is conserved among enterobacteria (*Figure 1A*). The response regulator NtrC is phosphorylated by its cognate sensor kinase NtrB at a low nitrogen state and activates RpoN (σ54)-dependent promoters by binding at specific enhancer-like sequences (*Ninfa and Magasanik, 1986*; *Ninfa et al., 1987*). The *glnALG* operon contains three promoters, *glnAp1*, *glnAp2*, and *glnLp*, and the proximal σ54-dependent promoter *glnAp2* is activated by NtrC when phosphorylated by NtrB in nitrogen-poor conditions while the other σ70-dependent promoters are repressed by NtrC (*Ueno-Nishio et al., 1984*; *Reitzer and Magasanik, 1985*).

The NtrB/NtrC system can be stimulated when the cells utilize Gln or low concentrations of ammonium as a sole nitrogen source in *E. coli* (*Schumacher et al., 2013*). Still, it has been postulated that a low intracellular Gln level is perceived as nitrogen limitation to induce the transcription of *glnA* in *S. enterica* (*Ikeda et al., 1996*). Alternatively, 2-OG has been proposed as a plausible signaling compound coordinating carbon and nitrogen metabolism since it serves as both the TCA cycle intermediary metabolite and the carbon skeleton of Glu and Gln (*Huergo and Dixon, 2015*). 2-OG accumulates rapidly upon nitrogen limitation from 0.2 to 10 mM (*Yuan et al., 2009*) and directly blocks glucose uptake by inhibiting enzyme I of the phosphoenolpyruvate phosphotransferase system (PTS) (*Doucette et al., 2011*). 2-OG also inhibits cAMP synthesis by adenylate cyclase (*You et al., 2013*). However, it is currently unknown how the cells adjust the 2-OG levels in response to nitrogen availability.

The genes under the direct transcriptional control of RpoN and NtrC have been extensively studied in *S. enterica* serovar Typhimurium (*Samuels et al., 2013*; *Bono et al., 2017*) and *E. coli* K-12 (*Brown et al., 2014*; *Zimmer et al., 2000*), but none of the genes are implicated in the central carbon metabolism. The regulons of alternative sigma factors and stress response regulators comprise coding and non-coding arms, the latter of which are mediated by small RNAs (*Hör et al., 2020*; *Nitzan et al., 2017*; *Beisel and Storz, 2010*; *Wagner and Romby, 2015*). The majority of sRNAs post-transcriptionally regulate target mRNAs through direct base-pairing *in trans*, which is facilitated by the RNA chaperone Hfq using multiple RNA-binding faces (*Vogel and Luisi, 2011*; *Updegrove et al., 2016*; *Kavita et al., 2018*). In addition to canonical sRNAs, 3′UTRs of mRNAs have emerged as a reservoir of post-transcriptional regulators (*Miyakoshi et al., 2015a*; *Adams and Storz, 2020*; *Ponath et al., 2022*). For example, the *gltIJKL* operon encoding the Glu/Asp ABC transporter is transcribed from σ70- and σ54-dependent promoters, and the *gltI* mRNA with a Rho-independent terminator is processed by RNase E to release SroC sRNA from its 3′UTR (*Miyakoshi et al., 2015b*). SroC functions as a sponge RNA of GcvB sRNA, which regulates >50 genes mostly involved in amino acid metabolism and transport (*Miyakoshi et al., 2022*). Likewise, the *glnA* mRNA contains a Rho-independent terminator with a moderate strength located upstream of *glnLp* (*Ueno-Nishio et al., 1984*), which can be a ligand of RNA chaperone Hfq (*Chen et al., 2019*). Indeed, an sRNA has been detected in the 3′UTR of *glnA* mRNA in *E. coli*, renamed as GlnZ (*Kawano et al., 2005*; *Walling et al., 2022*), and also in the same locus of *S. enterica* (*Sittka et al., 2009*). Therefore, GlnZ is regarded as a RpoN-dependent sRNA in charge of the non-coding arm of the NtrC regulon.

Here, we show that GlnZ derived from the 3′UTR of *glnA* mRNA represses the expression of *sucA* encoding the E1o component of 2-oxoglutarate dehydrogenase (OGDH) and thus functions as the

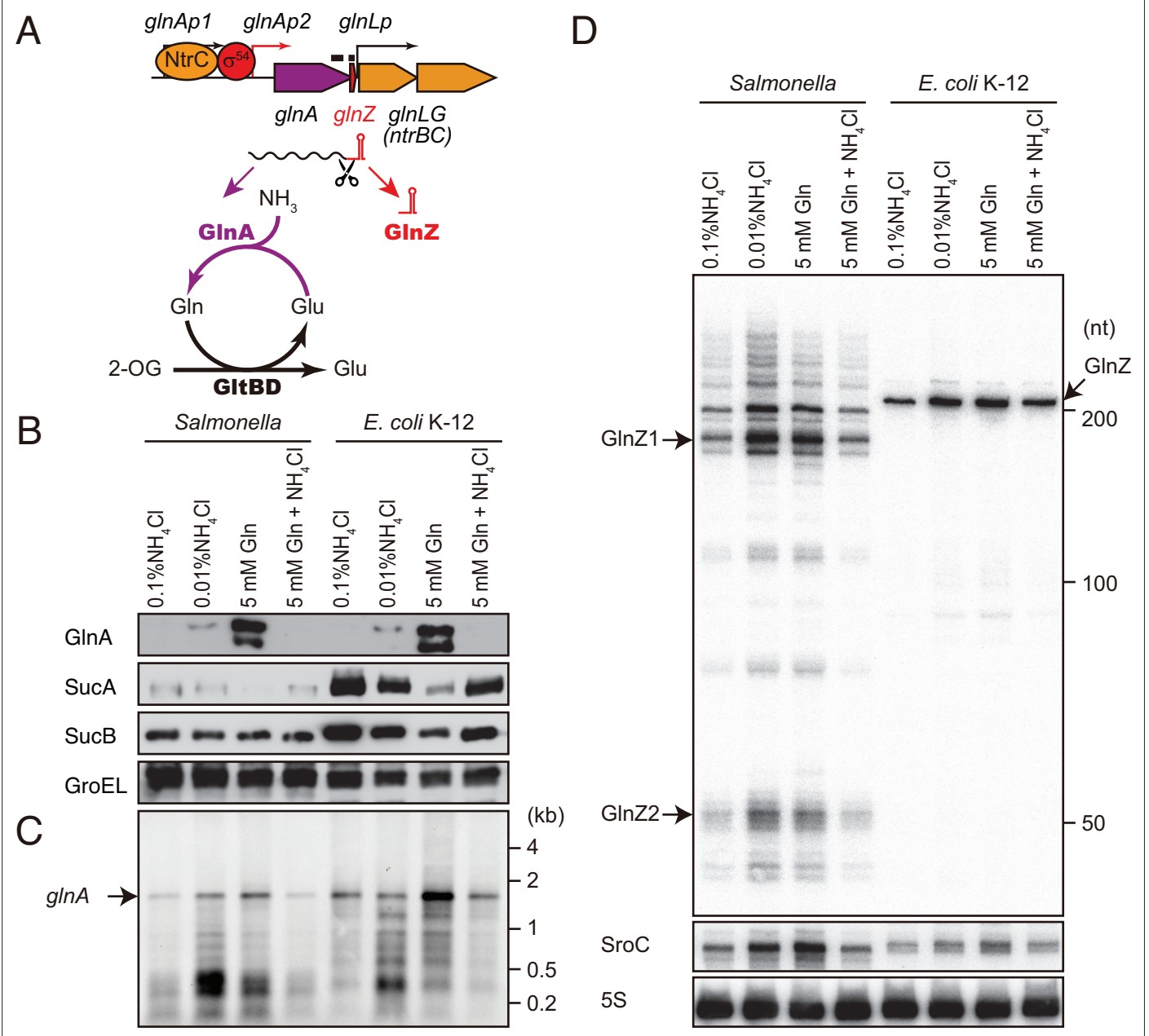

**Figure 1.** *glnA* mRNA expresses both GlnA protein and GlnZ sRNA. (**A**) The genetic structure of *glnALG* operon and nitrogen assimilation pathway. The transcription of *glnA* mRNA from σ⁵⁴-dependent promoter *glnAp2* is activated by the NtrB/NtrC two-component system, which is encoded downstream of *glnA*. GlnA (glutamine synthetase [GS]) catalyzes the amidation of glutamate (Glu) to form glutamine (Gln) using one molecule of ATP, and GltBD (glutamine 2-oxoglutarate amidotransferase [GOGAT]) transfers the amide group of Gln to 2-oxoglutarate (2-OG) to generate two Glu molecules. GlnZ is processed from the 3'UTR of *glnA* mRNA. The black bars above the genes indicate the location of probes used for northern blots. (**B**) Expression profiles of 2-oxoglutarate dehydrogenase (OGDH) subunits (SucA and SucB) and GlnA in *Salmonella enterica* and *Escherichia coli* during growth on different nitrogen sources. *S.* Typhimurium SL1344 and *E. coli* BW25113 were grown to exponential phase (OD₆₀₀ ~0.5) in MOPS media containing 0.2% glucose as the carbon source and the following nitrogen sources: 0.1% ammonium, 0.01% ammonium, 5 mM Gln, or 5 mM Gln plus 0.01% ammonium. SucA and SucB were detected by antibodies raised against purified *E. coli* proteins. GlnA was detected by an antibody raised against a synthetic peptide. GroEL served as a loading control. (**C and D**) Expression profiles of *glnA* mRNA and GlnZ sRNA. The *glnA* mRNA was detected by an equimolar mixture of two RNA probes specific for *Salmonella* and *E. coli*. GlnZ was detected by a common oligonucleotide probe MMO-0416. SroC sRNA was detected by an equimolar mixture of oligonucleotide probes JVO-2907 and JVO-5622. 5S rRNA detected by MMO-1056 served as a loading control.

The online version of this article includes the following source data for figure 1:

*Figure 1 continued on next page*

*Figure 1 continued*

**Source data 1.** Figure with the uncropped blots.

**Source data 2.** The original files of the full raw unedited northern blots.

**Source data 3.** The original files of the full raw unedited westen blots.

post-transcriptional regulator of the TCA cycle branch point directly linking carbon and nitrogen metabolism in *S. enterica* and *E. coli*. In parallel, the NtrC-dependent transcriptional regulator Nac also represses *sucA* in *E. coli* but is absent in *S. enterica*. Moreover, we demonstrate that RNase E-mediated cleavage of GlnZ from the *glnA* mRNA is essential for targeting the *sucA* mRNA.

## Results

### Expression profiles of OGDH subunits and the *glnA* products under nitrogen limitation

As 2-OG accumulates upon nitrogen limitation (*Yuan et al., 2009*), we hypothesized that the expression level of OGDH is reduced to create a bottleneck in the TCA cycle. To this end, we analyzed the expression levels of OGDH subunits, SucA and SucB, in *S*. Typhimurium SL1344 and *E. coli* K-12 by western blot. Throughout this study, we used 0.2% glucose MOPS minimal medium whose nitrogen source was substituted by either 0.1% ammonium, 0.01% ammonium, 5 mM Gln, or 5 mM Gln plus 0.01% ammonium. In line with a previous report (*Schumacher et al., 2013*), GlnA was induced when ammonium was limiting and to higher levels during growth on Gln in *S. enterica* and *E. coli* (*Figure 1B*). Addition of ammonium into the Gln medium hindered the induction, that is, nitrogen catabolite repression. In contrast, the SucA levels were strikingly reduced during growth on Gln and counter-correlated with the GlnA levels in both strains. In *E. coli*, SucB exhibited a similar expression pattern to SucA. However, SucB was expressed constantly under the four growth conditions in *S. enterica*. This result suggests that the expression of *sucAB* genes in the same operon is discoordinately regulated at the post-transcriptional level in response to nitrogen availability.

To clarify whether the levels of *glnA* mRNA and GlnZ sRNA correlate with protein accumulation, total RNAs were analyzed by northern blot. As expected, the *glnA* mRNA was induced during growth on Gln as the sole nitrogen source in *S. enterica* and *E. coli* (*Figure 1C*). At the low concentration of ammonium, degradation products in the 3′ region of *glnA* CDS accumulated as the cells starve for ammonium and stop growth. In the sRNA fractions from the *S. enterica* total RNA, we detected several GlnZ isoforms (*Figure 1D*). The size of the most abundant transcript, denoted as GlnZ1, matches that of the *glnA* 3′UTR spanning 185 nucleotides (nt), given the transcription stops at the Rho-independent terminator with 6 U residues. The transcripts longer than 200 nt represent precursors processed within the *glnA* CDS. We also detected ~50 nt short transcripts, denoted as GlnZ2, which are produced by RNase E-mediated cleavage (*Chao et al., 2017*). In contrast, an ~200 nt single transcript was seen in the *E. coli* K-12 total RNA using the same probe, which hybridizes the 5′ end region of K-12 GlnZ. In both *S. enterica* and *E. coli* K-12, the GlnZ transcripts were abundant under the two nitrogen limiting conditions. Similarly, we observed induction of *S. enterica* SroC in the same samples (*Figure 1D*). In contrast, the induction of SroC was only marginal in *E. coli* K-12 due to the IS*5* insertion upstream of *gltI* (*Zinser et al., 2003*).

### Sequence comparison of the *glnA-glnLG* intergenic region

As the fragment patterns of GlnZ transcripts were different between *S. enterica* and *E. coli*, we compared the sequences of *glnA* 3′UTRs in the *Enterobacteriaceae* family. The 185 nt *glnA* 3′UTR is >99% identical among *Salmonella* species/subspecies except for *S. enterica* subsp. *diarizonae* and *salamae* (*Figure 2A*). In contrast, *E. coli glnA* 3′UTRs are phylogenetically diverse and can be classified into at least three types. *E. coli* K-12 and several pathogenic strains harbor the 197 nt *glnA* 3′UTR (K-12 type). Many *E. coli* pathogenic strains represented by O157 carry the 85 nt 3′UTR (O157 type), and the others represented by O111 have the 230 nt 3′UTR (O111 type), the latter of which has acquired a 98 nt insertion flanked by 5′-AGGUUCAAA-3′ direct repeats into the O157-type *glnA* 3′UTR (*Figure 2B*). The *glnA* 3′UTRs of the other species in the *Escherichia* genus, *E. albertii* and *E. fergusonii*, are almost identical to the K-12 and O157 types, respectively.

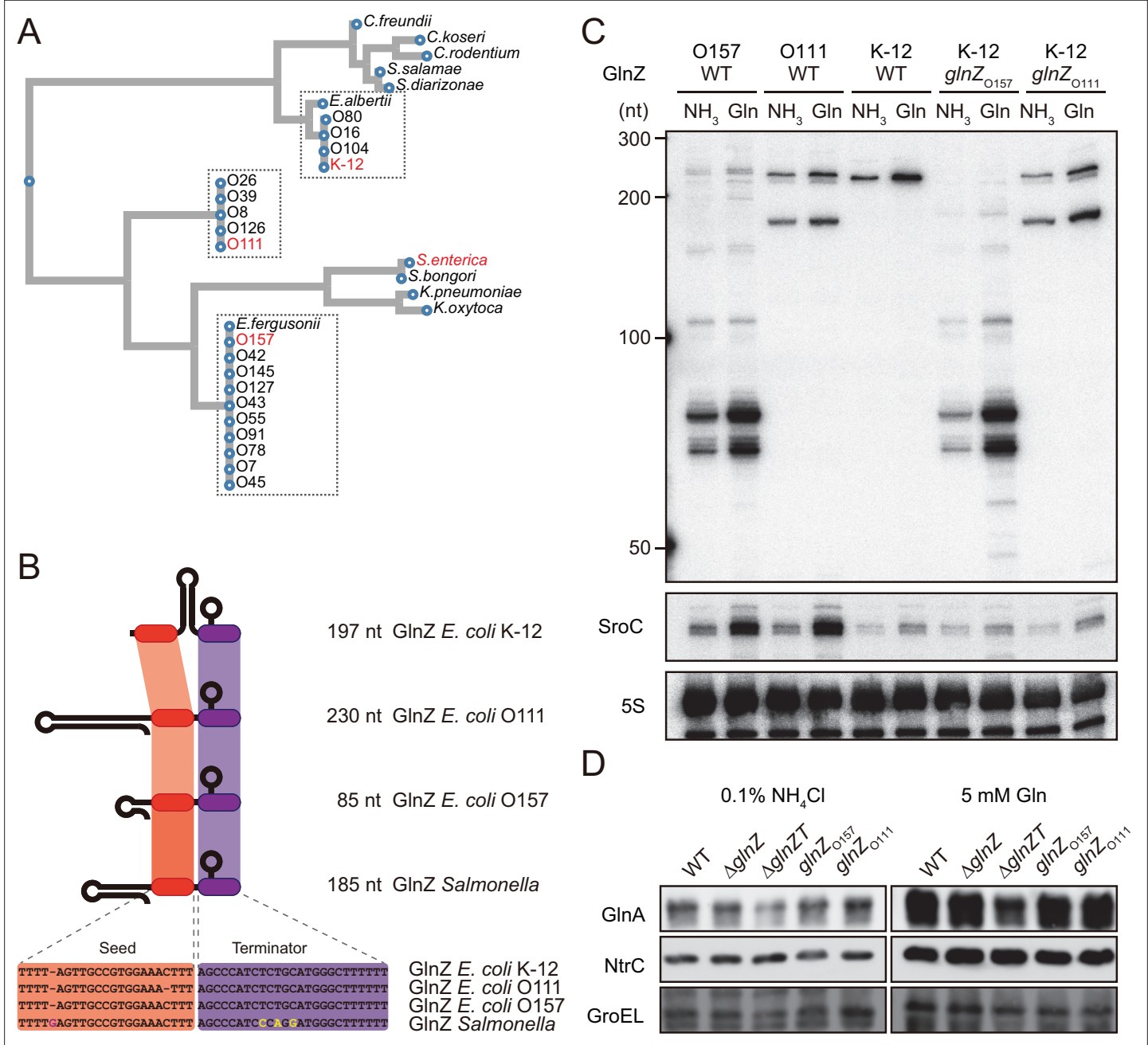

**Figure 2.** Variation of GlnZ sRNAs in the *Enterobacteriaceae* family. (**A**) Multiple sequence alignment of *glnA* 3'UTRs using CLUSTALW program (https://www.genome.jp/tools-bin/clustalw). (**B**) The location of conserved seed region and Rho-independent terminator in different types of GlnZ sRNAs. The extra G nucleotide and variable nucleotides found in the terminator of *Salmonella* GlnZ are indicated in purple and yellow letters. (**C**) *Escherichia coli* strains express different types of GlnZ sRNAs. *E. coli* O157, O111, and K-12 strains were grown to exponential phase ($OD_{600}$ ~0.5) in MOPS media containing 0.2% glucose as the carbon source and either 0.1% ammonium or 5 mM glutamine (Gln) as the nitrogen source. GlnZ and SroC sRNAs were detected by MMO-0416 and JVO-5622, respectively. 5S rRNA detected by MMO-1056 served as a loading control. (**D**) The difference in 3'UTR sequence does not affect the expression of downstream NtrC. *E. coli* K-12 strains, wild type (WT), Δ*glnZ*, Δ*glnZT*, *glnZ*$_{O157}$, and *glnZ*$_{O111}$, were grown to exponential phase ($OD_{600}$ ~0.5) in MOPS media containing 0.2% glucose as the carbon source and either 0.1% ammonium or 5 mM Gln as the nitrogen source. GlnA was detected by an antibody raised against a synthetic peptide. NtrC was chromosomally tagged with 3xFLAG and detected by α-FLAG antibody. GroEL served as a loading control.

The online version of this article includes the following source data for figure 2:

**Source data 1.** Figure with the uncropped blots.

**Source data 2.** The original files of the full raw unedited northern blots.

**Source data 3.** The original files of the full raw unedited western blots.

To verify whether these strains generate different types of GlnZ transcripts, total RNAs extracted from the O157 and O111 cells were analyzed by northern blot. We detected GlnZ sRNAs corresponding to the *glnA* 3′UTRs in size and their processed species, which were induced when Gln was used as the sole nitrogen source (*Figure 2C*). This result is consistent with the study by *Walling et al., 2022*, but not the previous report by *Jia et al., 2021*. In support, SroC was also induced under the nitrogen limiting condition in the two pathogenic *E. coli* strains devoid of IS5 insertion upstream of *gltI*.

The structures of *E. coli glnA* 3′UTRs might affect the expression of the downstream *glnLG* genes encoding NtrBC. In the K-12 strain, the expression of GlnA and NtrC was not affected by the partial deletion of *glnA* 3′UTR retaining the Rho-independent terminator (Δ*glnZ*) (*Figure 2D*). In contrast, the level of GlnA was slightly reduced by further deletion of the terminator (Δ*glnZT*) while that of NtrC remained nearly constant. These results suggest that the Rho-independent terminator stabilizes the *glnA* mRNA but not the *glnALG* mRNA or that the *glnLG* mRNA is transcribed independently to maintain the level of NtrC. Next, we replaced the *glnA* 3′UTR in the K-12 chromosome with that of O157 or O111. The recombinant strains expressed the heterologous GlnZs (*Figure 2C*), but the difference in the expression levels of GlnA and NtrC between these strains and the parental K-12 strain was negligible (*Figure 2D*). These results indicate that the various *glnALG* operons of *E. coli* strains are expressed equivalently.

## Identification of GlnZ targets in *S. enterica* and *E. coli*

Enterobacterial GlnZ sRNAs contain a conserved sequence, 5′-UUGCCGUGGAAA-3′, which is located adjacent to the Rho-independent terminator in *S. enterica* or just downstream of *glnA* stop codon in *E. coli* K-12. Thus, this sequence was assumed to function as the seed region to interact with target mRNAs (*Storz et al., 2011*; *Gorski et al., 2017*). We searched for the complementary sequences to the putative seed region in several enterobacterial genomes using the CopraRNA program (*Wright et al., 2013*). This prediction suggested that the primary target of GlnZ is most likely the *sucA* gene, which encodes the E1o subunit of OGDH. The *sucA* gene is located in the *sdhCDAB-sucABCD* operon, which is regulated by multiple sRNAs (*Desnoyers and Massé, 2012*; *Massé and Gottesman, 2002*; *Wright et al., 2013*; *Miyakoshi et al., 2022*) but also produces an sRNA regulator SdhX from its 3′UTR (*Miyakoshi et al., 2019*; *De Mets et al., 2019*). The *sdhB-sucA* intergenic region spans 523 nt in *S.* Typhimurium SL1344 and 300 nt in *E. coli* K-12. The sequence complementary to the GlnZ seed region is located upstream of the translation initiation region (TIR) of *sucA* in *E. coli* and far upstream in *S. enterica* (*Figure 3A and B*).

To verify the post-transcriptional regulation by GlnZ, the *sdhB-sucA* intergenic region was cloned into the pXG30-sf translational fusion vector (*Corcoran et al., 2012*). When either GlnZ1 or GlnZ2 was ectopically expressed from a constitutive promoter, the fluorescence of *S. enterica* SucA::sfGFP fusion was significantly reduced (*Figure 3A*). This result indicates that *sucA* is repressed by GlnZ1, and the shorter transcript GlnZ2 is sufficient to repress *sucA*. The repression was abrogated when the 149th nucleotide downstream of the *glnA* stop codon was mutated from G to C in GlnZ1 (G149C) and when the 166th nucleotide upstream of the *sucA* start codon was mutated from C to G in *sucA* (C-166G). The repression was restored when GlnZ1 and *sucA* were simultaneously mutated in the complementary nucleotides, demonstrating the post-transcriptional regulation through the base-pairing mechanism. Similarly, *E. coli* GlnZ repressed *sucA* irrespective of the position of seed region in the sRNA and that of target region in the mRNA (*Figure 3B*).

To search for additional targets of GlnZ, we performed RNA-seq analysis upon GlnZ1 pulse expression in *S. enterica* cells exponentially grown in LB medium. Among the 12 genes that showed more than fourfold downregulation (*Supplementary file 1*), *glnP* and *glnQ* constitute the *glnHPQ* operon encoding the Gln ABC transporter. We also found *deoD* encoding a purine nucleoside phosphorylase as a candidate GlnZ target. Using IntaRNA program (*Mann et al., 2017*), both *glnP* and *deoD* mRNAs were predicted to base-pair with the GlnZ seed region. Using the pXG30-sf constructs of *glnHP* and *deoBD* intergenic regions, we verified that the expression of *glnP* and *deoD* was repressed by *S. enterica* GlnZ1 and GlnZ2 through the base-pairing mechanism (*Figure 3C and D*). It is noteworthy that the *glnP* target region is located just upstream of the SD sequence and is conserved in *E. coli* as verified by the others (*Walling et al., 2022*). However, *Salmonella* GlnZ1 binds to the *deoD* mRNA at the first three codons, which are synonymously mutated in *E. coli*.

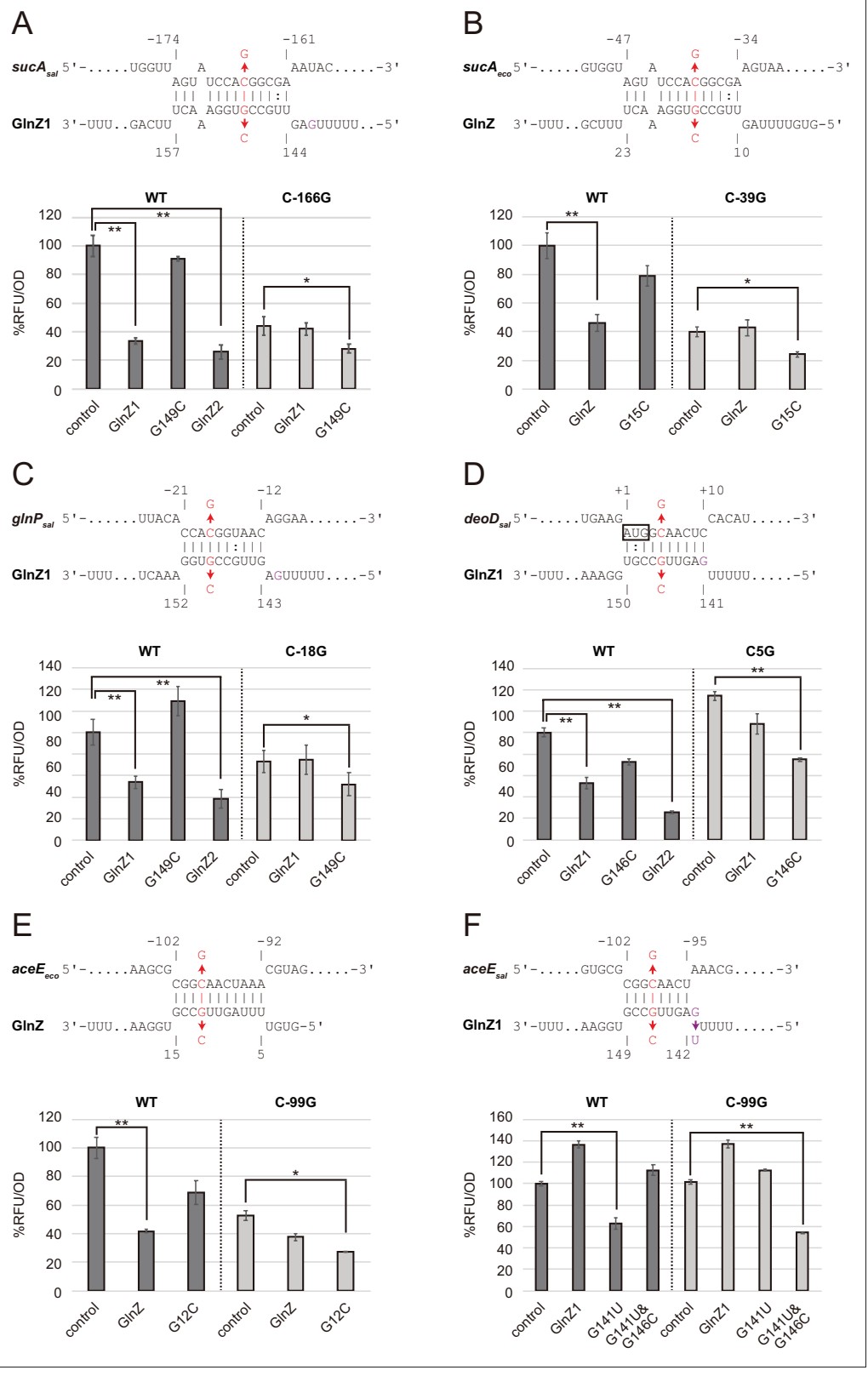

**Figure 3.** Post-transcriptional regulation mediated by *Salmonella enterica* and *Escherichia coli* GlnZ. Predicted interactions of GlnZ with the target mRNAs are shown in the panels, (**A**) *S. enterica sucA*, (**B**) *E. coli sucA*, (**C**) *S. enterica glnP*, (**D**) *S. enterica deoD*, (**E**) *E. coli aceE*, and (**F**) *S. enterica aceE*. The nucleotide numbers relative to the start codon of the target mRNA and the stop codon of *glnA* are shown above and below the

*Figure 3 continued on next page*

*Figure 3 continued*

nucleotide sequences, respectively. The mutated nucleotides are indicated in red, and the extra G nucleotide found in *Salmonella* GlnZ1 is shown in purple. *E. coli* Δ*glnZ* strain was transformed by GFP translational fusion plasmids along with pJV300 control vector or GlnZ expression plasmids (**Supplementary file 5-7**). Mean relative fluorescence units (RFU) normalized by $OD_{600}$ calculated from biological replicates (n>3) are presented with standard deviation in percentage relative to the vector control. Statistical significance was calculated using one-way ANOVA and denoted as follows: **p<0.005, *p<0.05.

The online version of this article includes the following figure supplement(s) for figure 3:

**Figure supplement 1.** Post-transcriptional regulation of *sucA* involves a purine-rich sequence.

**Figure supplement 2.** Post-transcriptional regulation of *aceE* by $GlnZ_{O157}$.

Next, we looked into the RNA-RNA interactome datasets in *E. coli* K-12 and UPEC O127:H6 strains (**Melamed et al., 2016**; **Melamed et al., 2020**; **Pearl Mizrahi et al., 2021**). In the cells grown in LB medium, the majority of GlnZ transcripts interacted with the *sdhB-sucA* intergenic region. Interestingly, GlnZ was also found to associate with the *pdhR-aceE* intergenic region. Using the pXG30-sf translational fusion of *pdhR-aceE*, we observed significant repression of AceE::sfGFP fusion by *E. coli* GlnZ (**Figure 3E**). The repression was relieved by the G12C mutation in GlnZ, but was restored by its complementary C-99G mutation in *aceE*, showing that GlnZ regulates *aceE* through base-pairing far upstream of TIR. However, the expression of *aceE* was not repressed by *S. enterica* GlnZ1 (**Figure 3F**). We reasoned that the seed region of GlnZ1 contains G instead of U at the 143rd nucleotide, and thus the interaction with *aceE* is weaker than the *E. coli* GlnZ. When the nucleotide is substituted by U (G143U), GlnZ1 gained the inhibitory effect on *S. enterica aceE* by extending the hybridization (**Figure 3D**). Overall, GlnZ post-transcriptionally regulates *sucA* and *glnP* in common, but the mutations adjacent to the seed region and in the target brought about the species-specific regulation of *aceE* and *deoD*, respectively.

## GlnZ represses SucA during growth on Gln as the sole nitrogen source

As the post-transcriptional regulation by GlnZ through the base-pairing mechanism has been verified, we evaluated the effect of endogenous GlnZ on its primary target *sucA*. To this end, we constructed mutant strains partially lacking the *glnA* 3′UTRs (Δ*glnZ*) of *S.* Typhimurium SL1344 and *E. coli* K-12 and analyzed the SucA levels by western blot. These deletion mutants grew in the minimal media similarly to the respective wild-type (WT) strains. The levels of SucA protein were twice as high during growth in ammonium-containing media than during growth on Gln as the sole source of nitrogen but were not significantly altered by the deletion of *glnZ* (**Figure 4A**). However, the *S. enterica* Δ*glnZ* strain expressed ~1.3-fold higher levels of SucA than WT during growth on Gln (**Figure 4A**), indicating that GlnZ represses SucA when the basal expression level of SucA is low.

Likewise, we observed a modest but significant increase (~1.4-fold) in the SucA expression level in the *E. coli glnZ* mutant during growth on Gln (**Figure 4B**). Still, the difference was much greater (~5-fold) when compared between the poor and rich nitrogen sources, implicating additional regulators in the control of SucA expression. Recently, the global regulator of nitrogen assimilation control (Nac), which is absent in *S. enterica* (**Muse and Bender, 1998**), was found to bind the *sucA* promoter in *E. coli* (**Aquino et al., 2017**). In line with this finding, SucA and SucB exhibited a similar expression pattern during growth with different nitrogen sources in *E. coli*, while the level of SucB was constant in *S. enterica* (**Figure 1B**). We verified that the deletion of *nac* alone slightly increased the SucA level without affecting the GlnZ level, but the double mutant of *nac* and *glnZ* further raised the SucA level by ~2.5-fold compared to WT (**Figure 4B**). As Nac and GlnZ are induced through transcriptional activation by NtrC, *E. coli* utilizes the transcriptional and post-transcriptional regulators to repress the same target in parallel.

## Hfq is required for GlnZ biogenesis and its target regulation

Since *E. coli* GlnZ is classified as a class I sRNA which binds to proximal and rim faces of RNA chaperone Hfq (**Kavita et al., 2022**), Hfq is implicated in the stability of GlnZ. Using the probe that hybridizes the 3′ end region of K-12 GlnZ, we detected a few bands of GlnZ, denoted as $GlnZ_{213}$, $GlnZ_{194}$, and $GlnZ_{174}$ by the others (**Walling et al., 2022**). In the *hfq* mutant, the *glnA* mRNA was induced under the nitrogen limiting conditions, but we detected only faint fragments of $GlnZ_{174}$ (**Figure 5A**). This result

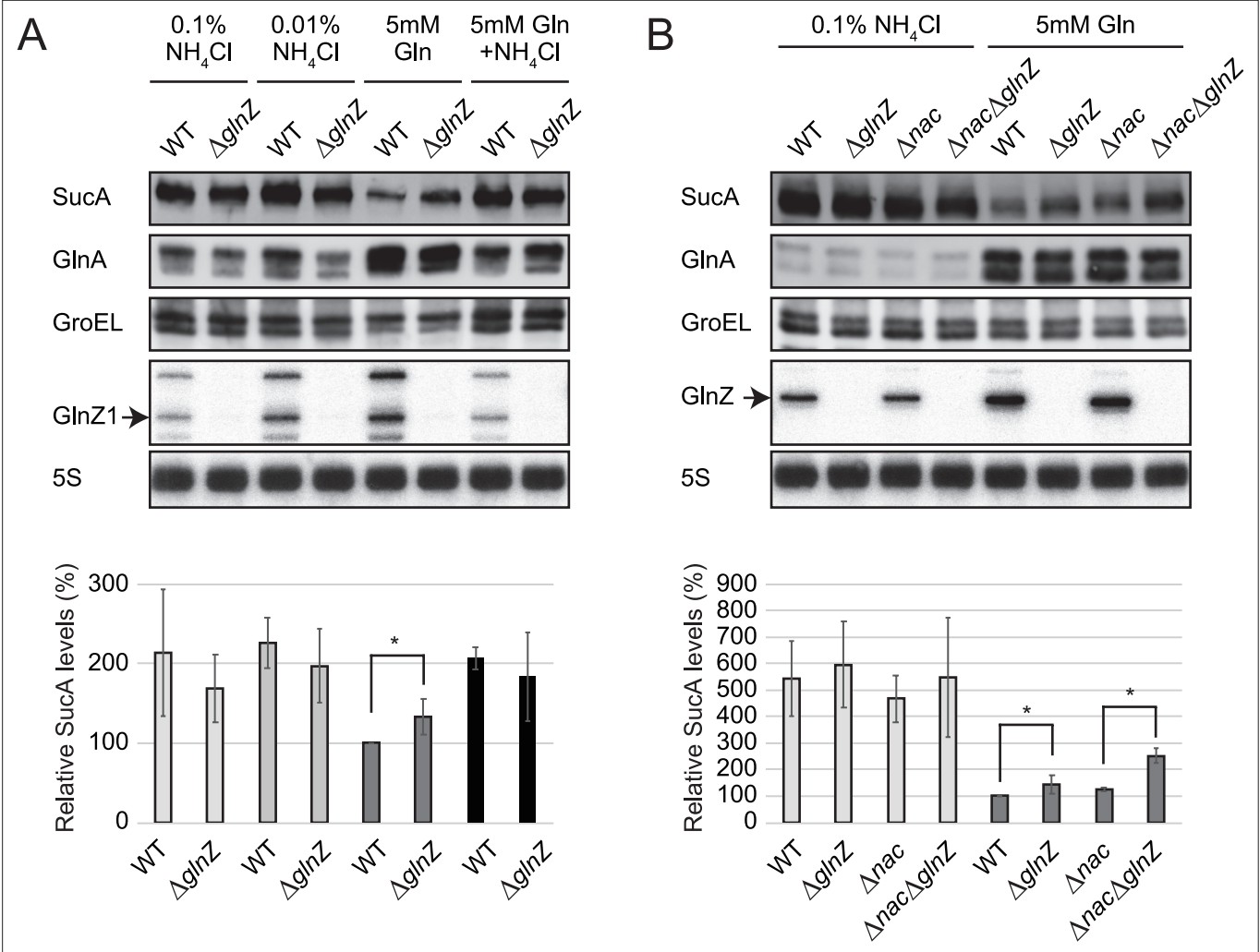

**Figure 4.** Repression of SucA by endogenous GlnZ. (**A**) GlnZ post-transcriptionally represses the expression of SucA in *Salmonella* during growth on glutamine (Gln) as the nitrogen source. *Salmonella* wild-type (WT) and Δ*glnZ* strains were grown to exponential phase (OD$_{600}$ ~0.5) in MOPS minimal medium containing 0.2% glucose as the carbon source and different nitrogen sources; 0.1% ammonium, 0.01% ammonium, 5 mM Gln, and 5 mM Gln plus 0.01% ammonium. (**B**) GlnZ and Nac independently repress the expression of SucA in *Escherichia coli* during growth on Gln as the nitrogen source. *E. coli* WT, Δ*glnZ*, Δ*nac*, and Δ*glnZ*Δ*nac* strains were grown to exponential phase (OD$_{600}$ ~0.5) in MOPS minimal medium containing 0.2% glucose as the carbon source and either 0.1% ammonium or 5 mM Gln as the nitrogen source. The expression of SucA and glutamine synthetase (GS) was analyzed by western blots (upper panels), and that of GlnZ was analyzed by northern blots (bottom panels). GroEL served as a loading control for western blots; 5S rRNA for northern blots. The expression levels of SucA relative to GroEL were normalized to that of WT strain grown on Gln, and the standard deviation was calculated from biological replicates (n>3). Statistical significance was calculated using one-way ANOVA and denoted as *p<0.05.

The online version of this article includes the following source data for figure 4:

**Source data 1.** Figure with the uncropped blots.

**Source data 2.** The original files of the full raw unedited northern blots.

shows that Hfq is essential for GlnZ biogenesis. Interestingly, the levels of GlnA protein were rather elevated in the absence of Hfq, implying that the expression of GlnA is subject to post-transcriptional regulation by Hfq.

To clarify the reason why GlnZ was not generated in the *hfq* mutant, GlnZ was ectopically expressed as a primary transcript as in the reporter assays. GlnZ and its mutants, G15C and G12C, were equally abundant in the *E. coli* Δ*glnZ* strain grown exponentially in LB medium. In the *hfq* mutant, we detected ectopically expressed GlnZ though at lower levels than in WT (*Figure 5B*), indicating that Hfq stabilizes the GlnZ transcript. Concomitantly, the expression of target genes was analyzed by western blot. The level of SucA was significantly reduced by GlnZ but not by the G15C mutant (*Figure 5B*).

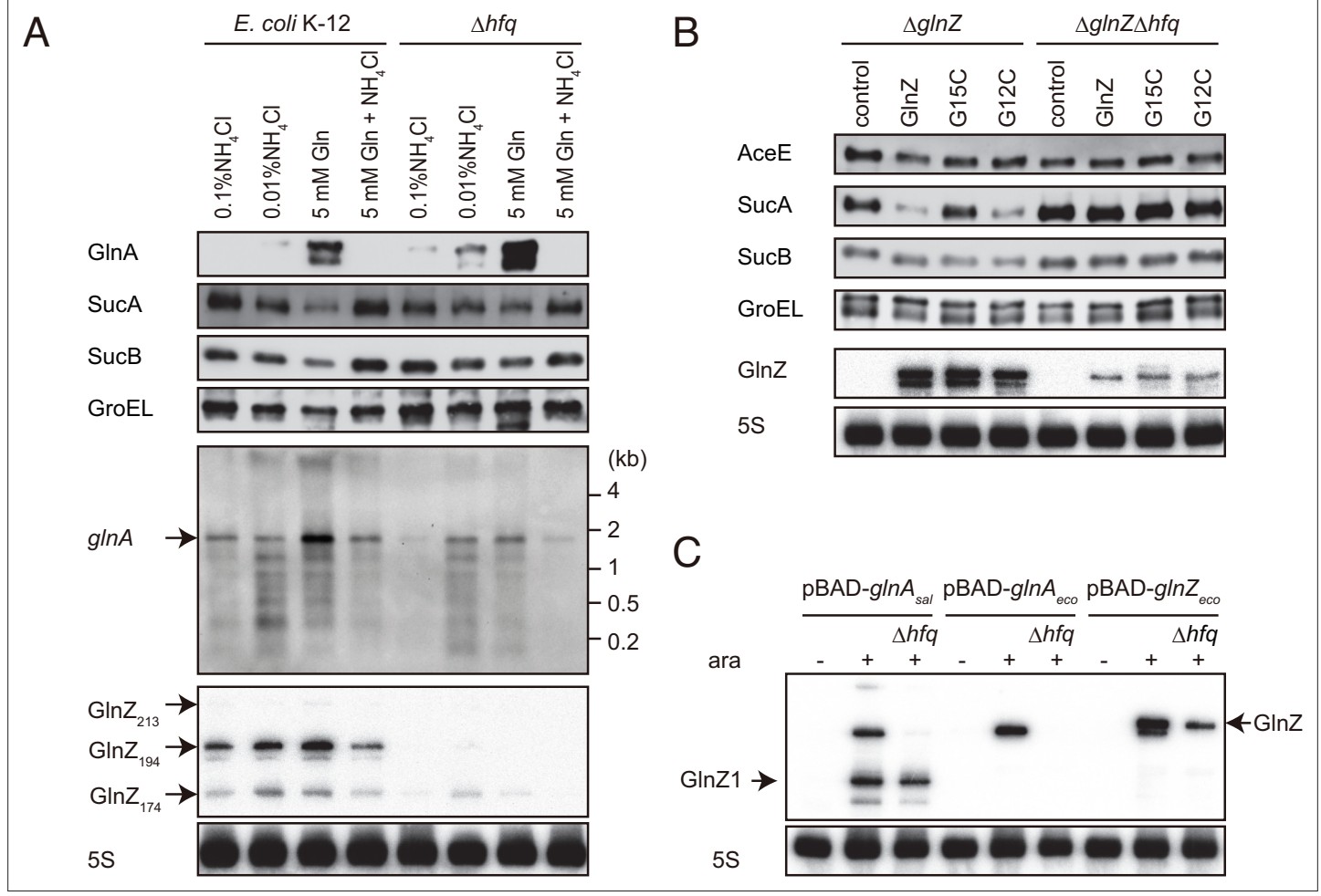

**Figure 5.** Hfq is essential for GlnZ biogenesis and target regulation. (**A**) Expression profiles of *glnA* gene products in *Escherichia coli* and its *hfq* mutant during growth on different nitrogen sources. *E. coli* BW25113 and its *hfq* mutant were grown to exponential phase (OD$_{600}$ ~0.5) in MOPS media containing 0.2% glucose as the carbon source and the following nitrogen sources: 0.1% ammonium, 0.01% ammonium, 5 mM Gln, or 5 mM Gln plus 0.01% ammonium. The protein and RNA levels were analyzed as in *Figure 1BCD* except *E. coli* GlnZ detection by a specific oligonucleotide probe MMO-0419. (**B**) Target regulation by ectopic expression of GlnZ and its derivatives. *E. coli* Δ*glnZ* and Δ*glnZ*Δ*hfq* strains harboring the constitutive GlnZ expression plasmids (*Supplementary file 5*) were grown to exponential phase (OD$_{600}$ ~1.0) in LB medium. The expression of AceE, SucA, and SucB was analyzed by western blots (upper panels), and that of GlnZ was analyzed by northern blots (bottom panels). GroEL served as a loading control for western blots; 5S rRNA for northern blots. (**C**) GlnZ is generated primarily from the *glnA* mRNA. *E. coli* Δ*glnZ* and Δ*glnZ*Δ*hfq* strains harboring pBAD-*glnA*$_{sal}$, pBAD-*glnA*$_{eco}$, or pBAD-GlnZ$_{eco}$ were grown to exponential phase (OD$_{600}$ ~1.0) at 37°C and further incubated for 10 min in the presence (+) or absence (-) of 0.2% L-arabinose.

The online version of this article includes the following source data for figure 5:

**Source data 1.** Figure with the uncropped blots.

**Source data 2.** The original files of the full raw unedited western blots.

**Source data 3.** Figure with the uncropped blots.

**Source data 4.** The original files of the full raw unedited northern blots.

**Source data 5.** Figure with the uncropped blots.

**Source data 6.** The original files of the full raw unedited western blots.

**Source data 7.** Figure with the uncropped blots.

**Source data 8.** The original files of the full raw unedited western blots.

The G12C mutation did not significantly affect the regulation of *sucA* in line with the predicted base-pairing interaction (*Figure 3B*). In contrast, the level of SucB was unaffected even if GlnZ was over-expressed, showing that GlnZ-mediated repression does not extend to the downstream gene. GlnZ also repressed the chromosomally expressed AceE protein, but either G15C or G12C substitution modestly relieved the repression in line with the predicted base-pairing interaction (*Figure 3E*). These results suggest that Hfq is involved in the stability of GlnZ and the target regulation by GlnZ.

Notably in *S. enterica*, the binding site of GlnZ is located far upstream of the *sucA* TIR, where GlnZ is unlikely to compete with 30S ribosomes directly (*Fröhlich and Papenfort, 2020*). We found a purine-rich sequence in the vicinity of *sucA* TIR in both *E. coli* and *S. enterica* (*Figure 3—figure supplement 1A*). This is reminiscent of the noncanonical mechanism of Hfq-mediated repression of *sdhC*, in which a class I sRNA Spot42 binds far upstream of TIR and recruits Hfq to the TIR-proximal region to inhibit translation directly (*Desnoyers and Massé, 2012*). To clarify the role of the purine-rich sequence of *sucA*, we deleted the 15 nt region from the translational fusion constructs of *S. enterica* and *E. coli sucA* (Δ15). For *S. enterica sucA*, the longer spacer region was also deleted (Δ127 and Δ142) to make the distance between the target site and TIR equal to the *E. coli sucA* counterparts (*Figure 3—figure supplement 1A*). The deletion of the 15 nt region resulted in a significant decrease of the basal translation levels of *S. enterica* and *E. coli sucA* (*Figure 3—figure supplement 1B*). However, GlnZ had no significant effect on the translational fusion lacking the 15 nt region even though the seed region got closer to the TIR. These results show that the purine-rich sequence upstream of TIR is critical for the GlnZ-mediated repression of *sucA* translation.

## Hfq and RNase E are indispensable for GlnZ processing

A 3'UTR-derived sRNA is either transcribed from an ORF-internal promoter as a primary transcript or released from an mRNA by a ribonuclease (*Miyakoshi et al., 2015a*). To clarify how GlnZ is generated, the *glnA* genes of *S. enterica* and *E. coli* were ectopically expressed from an arabinose-inducible P$_{BAD}$ promoter in the *E. coli* Δ*glnZ* background. With this plasmid expression system, we detected processed fragments as in the endogenous transcripts but virtually no transcripts in the absence of L-arabinose (*Figure 5C*), confirming that GlnZ is generated from the *glnA* mRNA. In the *hfq* mutant, *E. coli* GlnZ was successfully expressed as a primary transcript but not when transcribed along with the *glnA* mRNA from the P$_{BAD}$ promoter (*Figure 5C*). This result suggests that Hfq is required for the processing of *E. coli* GlnZ. However, the *S. enterica glnA* mRNA produced somewhat different fragments in the absence of Hfq; although a precursor processed upstream of the *glnA* stop codon was not detected, a substantial amount of GlnZ1 was still produced.

As many 3'UTR-derived sRNAs are processed by RNase E (*Ponath et al., 2022*), several RNase E cleavage sites have been identified in the *glnA* 3'UTR in *S. enterica* (*Chao et al., 2017*). To clarify the requirement of RNase E, the *glnA* mRNA of *S. enterica* or *E. coli* was pulse-expressed for 10 min either in the *E. coli* WT or temperature-sensitive RNase E mutant *ams-1* (TS). The fragment patterns were comparable between WT and TS strains at the permissive temperature. At the non-permissive temperature, the TS strain exhibited no accumulation of processed GlnZ sRNAs (*Figure 5B*). Thus, we conclude that RNase E is critical for GlnZ biogenesis from the *glnA* mRNAs.

## sRNA release from its parental mRNA is essential for target regulation

GlnZ is crucial for the post-transcriptional regulation of *sucA*. Still, the parental *glnA* mRNA can also be the regulator because it contains the seed region and Rho-independent terminator, both of which are a prerequisite for Hfq-mediated interaction with a target mRNA. To this end, we analyzed whether GlnZ processing is required for its target regulation using the *E. coli* K-12 pBAD-*glnA* variants. We identified two RNase E cleavage sites flanking the *glnA* stop codon, which are conformable to the consensus motif (*Chao et al., 2017*), and introduced mutations into the pBAD-*glnA*$_{eco}$ plasmid to disrupt the processing by RNase E without affecting the seed region (*Figure 6B*). The mutation at the downstream site (mut1) strongly inhibited the processing but slightly stimulated the cleavage at the upstream site compared to WT while the mutation at the upstream site (mut2) did not affect the downstream cleavage (*Figure 6C*). Simultaneous mutations at the two cleavage sites (mut3) almost completely abrogated the processing of the *glnA* mRNA (*Figure 6C*). This result suggests that RNase E processively cleaves the two sites of the *E. coli glnA* mRNA in the 5' to 3' direction. It is noteworthy

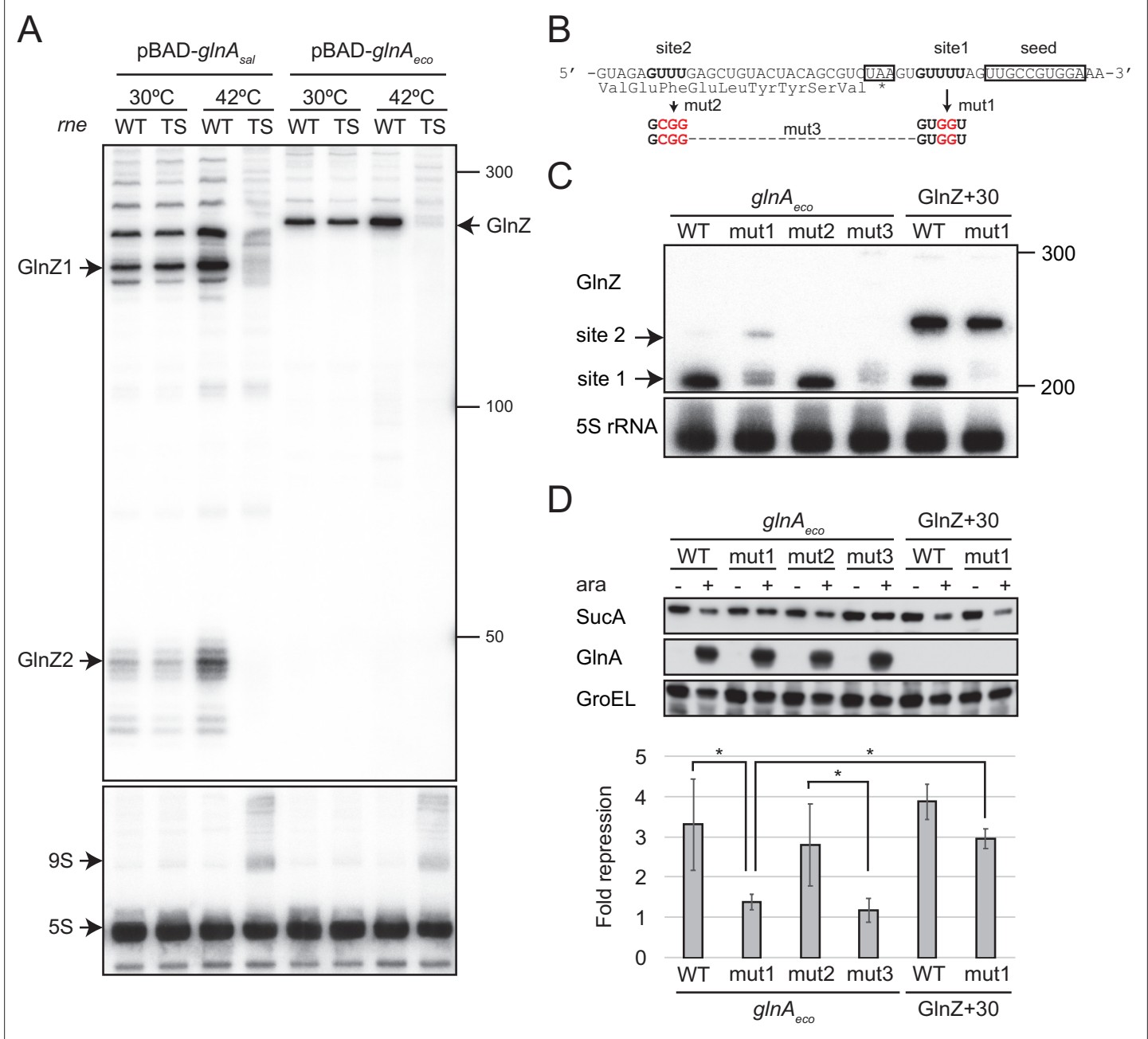

**Figure 6.** GlnZ release from the *glnA* mRNA is necessary for target repression. (**A**) RNase E is essential for the processing of *glnA* mRNA. *Escherichia coli* Δ*glnZ* (wild-type [WT]) and Δ*glnZ ams*-1 (TS) strains harboring pBAD-*glnA*sal or pBAD-*glnA*eco were grown to OD600 ~0.5 at 30°C and split into two flasks. The flasks were incubated at either 30°C or 42°C for 30 min and further incubated for 10 min after adding 0.2% L-arabinose. The size is estimated by DynaMarker RNA Low II ssRNA fragment. (**B**) The nucleotide sequence of RNase E cleavage sites in *E. coli* K-12 *glnA* mRNA. The C-terminal amino acid sequence of GlnA is shown below the nucleotide sequence. The *glnA* stop codon and the GlnZ seed region are boxed. The mutated nucleotides are indicated in red. (**C**) The processing of GlnZ is abrogated by the mutations in RNase E cleavage sites. GlnZ processed from either the *glnA* mRNA or the GlnZ precursor sRNA was analyzed by northern blot. 5S rRNA served as a loading control. (**D**) The processing of GlnZ is required for the repression of SucA in the form of mRNA but not the precursor sRNA. Expression levels of SucA and GlnA were analyzed by western blot. GroEL served as a loading control. *E. coli* Δ*glnZ* strains harboring pBAD expression plasmids were grown to exponential phase (OD$_{600}$ ~1.0) in LB medium in the absence (-) or presence of 0.01% L-arabinose (+). Bar graph represents the fold change of SucA repression by the *glnA* mRNA induced by arabinose calculated from biological replicates (n>5) with standard deviation. Statistical significance was calculated using one-way ANOVA and denoted as *p<0.05.

The online version of this article includes the following source data and figure supplement(s) for figure 6:

**Source data 1.** Figure with the uncropped blots.

*Figure 6 continued on next page*

*Figure 6 continued*

**Source data 2.** The original files of the full raw unedited northern blots.

**Source data 3.** Figure with the uncropped blots.

**Source data 4.** The original files of the full raw unedited northern and western blots.

**Figure supplement 1.** Expression of GlnZ and its targets in *Escherichia coli rnc14* mutant.

that the upstream RNase E cleavage site is conserved in *S. enterica glnA* and may be responsible for the processing of the ~200 nt precursor (**Figures 5C and 6A**).

Next, we analyzed the protein levels upon ectopic expression of the pBAD-*glnA*$_{eco}$ variants. Western blot analysis showed that these plasmids expressed GlnA at strikingly higher levels than the endogenous level (**Figure 6D**). This result also confirms that mutations in RNase E cleavage sites did not affect the translation of GlnA. Ectopic expression of the WT *glnA*$_{eco}$ mRNA significantly repressed SucA, but when mut1 was introduced into the *glnA* mRNA, SucA expression was no longer affected (**Figure 6D**). Notably, this mutation also abrogated the processing of a precursor GlnZ with the 30 nt extension at the 5′ end (GlnZ + 30) but still repressed the expression of SucA (**Figure 6C and D**). This result indicates that GlnZ needs to be separated from its parental mRNA to regulate its target mRNA.

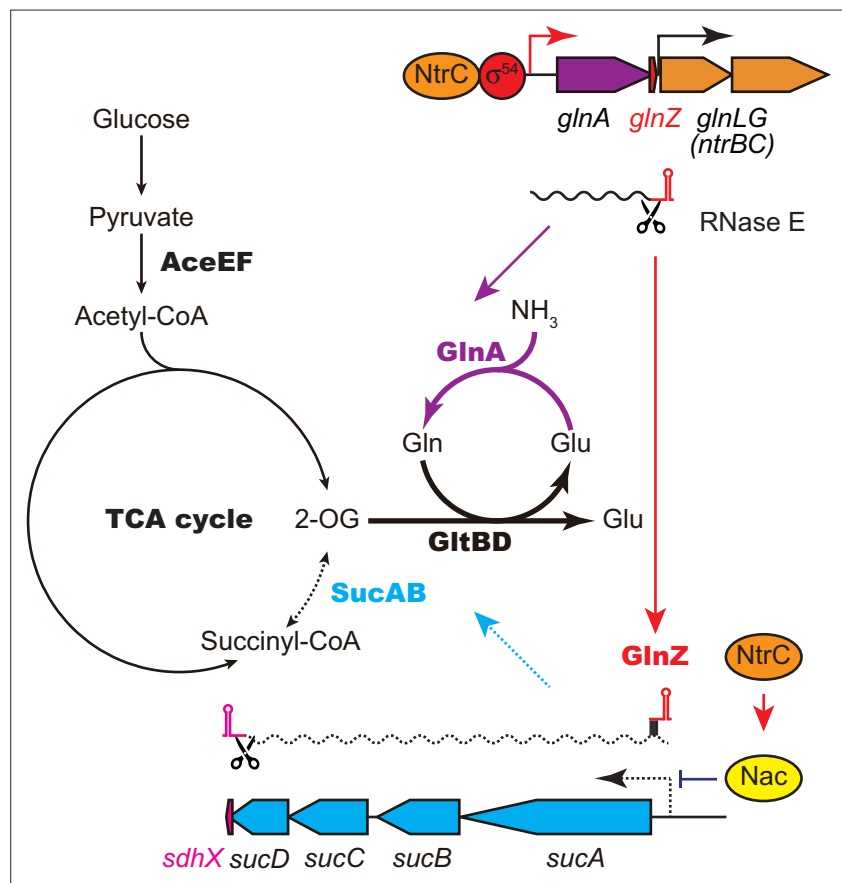

**Figure 7.** GlnA and GlnZ expressed from *glnA* mRNA facilitate nitrogen assimilation independently. The expression of GlnA and GlnZ is induced by the two-component system NtrBC upon nitrogen limitation. The glutamine synthetase (GS)-glutamine 2-oxoglutarate amidotransferase (GOGAT) (GlnA-GlnBD) pathway assimilates ammonia using 2-oxoglutarate (2-OG) as the carbon skeleton. GlnZ sRNA is released from the 3′UTR of *glnA* mRNA by RNase E to repress the expression of SucA at the post-transcriptional level. In *Escherichia coli*, the transcriptional regulator nitrogen assimilation control (Nac), whose expression is activated by NtrC, represses the transcription of *sucABCD* operon in parallel. The repression of 2-oxoglutarate dehydrogenase (OGDH) (SucAB) upon nitrogen limitation results in the accumulation of 2-OG and redirects the carbon flow from the TCA cycle to the GS-GOGAT pathway.

If the *glnA* mRNA is not processed properly, the translation might interfere with the base-pairing interaction with the target mRNA.

## Discussion

This study shows that the expression of SucA is repressed in *S. enterica* and *E. coli* during growth on Gln while the expression of GS is strikingly induced (*Figure 1B*), which leads to an increased metabolic flux of 2-OG from the TCA cycle to the GS-GOGAT pathway. Through the transcriptional activation of *glnA* by the NtrBC two-component system, GlnZ concomitantly represses the *sucA* mRNA at the post-transcriptional level (*Figure 7*). Thus, both the sRNA and protein expressed from the same mRNA act independently to balance the supply and demand of the fundamental intermediates of carbon and nitrogen metabolism. This study also highlights a difference between *S. enterica* and *E. coli*. The NtrBC-activated transcriptional regulator Nac reinforces the repression of *suc* operon at the transcriptional level in parallel in *E. coli*, whereas *S. enterica* devoid of *nac* exhibits discoordinated expression of SucA and SucB in response to nitrogen availability. Moreover, the position of seed region and the length of GlnZ are variable among *E. coli* species, and a single nucleotide substitution alters the specificity of target mRNAs. Importantly, GlnZ is primarily generated by RNase E-dependent cleavage from the 3′UTR of *glnA* mRNA. Although the *glnA* mRNA possesses the regulatory module within its 3′UTR, the mRNA devoid of processing is unable to regulate its target mRNA *in trans*. Thus, the sRNA needs to be separated from its parental mRNA to regulate its target mRNA.

### Conservation and variation of the *glnAZLG* locus and the GlnZ regulon

The *E. coli glnA* 3′UTR can be classified into three classes. We demonstrate that the difference in the *glnA* 3′UTRs does not affect the expression of downstream *glnLG* genes (*Figure 2D*), and thus the regulon of NtrBC two-component system. The *glnAZLG* genetic organization is conserved in *Enterobacteriaceae*, but *glnA* and *glnLG* (*ntrBC*) genes are often separated in the other γ-Proteobacteria. For instance, *Pseudomonas aeruginosa* harbors four ORFs between *glnA* and *ntrBC*. Moreover, a RpoN-dependent sRNA NrsZ has been found downstream of *ntrBC*, which regulates rhamnolipid biosynthesis and motility (*Wenner et al., 2014*). In the corresponding locus, *Salmonella* carries the *yshB* gene (*Hemm et al., 2008*), which encodes a small protein involved in the intracellular replication of *Salmonella* (*Bomjan et al., 2019*). However, *yshB* contains its own promoter and is not induced in nitrogen limiting conditions.

The GlnZ seed region is highly conserved to regulate *sucA* and *glnP* in common but is located at different positions in the *glnA* 3′UTR in *S. enterica* and *E. coli*. The repression of GlnHPQ ABC transporter by GlnZ may form a negative feedback loop to maintain the intracellular concentration of Gln. Each GlnZ might regulate other species-specific genes using its unique sequence, as in the case of *E. coli* SdhX repressing *katG* (*De Mets et al., 2019*; *Miyakoshi et al., 2019*).

By contrast, *aceE* is repressed by GlnZ in *E. coli*, but *S. enterica* GlnZ1 is deficient in this regulation due to the extra G adjacent to the seed region (*Figure 3E and F*). We further identified the *S. enterica*-specific target *deoD* encoding the purine nucleoside phosphorylase, with which base-pairing interaction involves the extra G nucleotide of *S. enterica* GlnZ1 (*Figure 3D*). Notably, none of the *E. coli* strains have the G nucleotide adjacent to the seed region, but 11 out of 55 *E. fergusonii* strains possess the 41st G nucleotide instead of U in the *glnA* 3′UTRs (*Figure 3—figure supplement 2*). *E. fergusonii* GlnZ resembles that of *E. coli* O157 with a few nucleotide substitutions, and as expected, the U41G mutation in the O157 GlnZ reduced its activity on *aceE* (*Figure 3—figure supplement 2*). We speculate that the base-pairing between *pdhR-aceEF* mRNA and GlnZ is beneficial for *E. coli* but is subject to trade-offs among the other targets.

Both *sucA* and *aceE* mRNAs are targeted by GlnZ in *E. coli* and encode components of analogous multienzyme complexes, OGDH and PDH, respectively. These enzymes associate with the common E3 subunit, Lpd, and are competitive for the cofactor HS-CoA to produce succinyl-CoA and acetyl-CoA, respectively (*Shimada et al., 2021*). Moreover, these target mRNAs contain relatively long stem-loop structures recognized by RNase III in *E. coli* (*Cunningham and Guest, 1998*; *Gordon et al., 2017*). Intriguingly, we found that SucA was downregulated and AceE was upregulated in the *rnc14* mutant while SucB in the same operon was unaffected (*Figure 6—figure supplement 1*), showing specific RNase III-mediated regulation of the two genes in opposite directions. RNase III cleaves the *sucA*

5′UTR at three sites upstream of the GlnZ target site (*Cunningham and Guest, 1998*), but the RNase III cleavage site on the 5′UTR of *aceE* mRNA is located downstream of the GlnZ target site (*Gordon et al., 2017*). Thus, the *aceE* mRNA becomes resistant to GlnZ if fully processed by RNase III. Our translational fusion constitutively transcribing the *pdhR-aceE* intergenic region was sensitive to GlnZ probably due to the high levels of ectopic expression (*Figure 4E*).

It has recently been reported that GlnZ of *E. coli* K-12 is also a substrate of RNase III *in vitro* as it contains a long stem-loop structure (*Walling et al., 2022*). We tested whether the processing of *glnA* mRNA is modified in the *rnc14* mutant using the pBAD expression system. However, we observed no significant changes in the expression patterns of GlnZ from the *glnA* mRNA (*Figure 6—figure supplement 1*), suggesting that RNase III plays a minor role in the processing of GlnZ. However, Walling et al. argue that RNase III is involved in the target regulation upon GlnZ overexpression (*Walling et al., 2022*). Further investigation is necessary to elucidate how RNase III, RNase E, and Hfq mediate the interplay between these important metabolic mRNAs.

## Difference between mRNA-derived sRNAs and dual-function sRNAs

mRNA-derived sRNAs have emerged from diverse regions of mRNAs (*Adams and Storz, 2020*). In enterobacteria, 3′UTR often binds with RNA chaperones Hfq and ProQ, thus serving as a reservoir of functional sRNAs (*Miyakoshi et al., 2015a*; *Ponath et al., 2022*). Surprisingly, regardless of Hfq functionality, 3′UTR-derived sRNAs are also found in Gram-positive bacteria (*Fuchs et al., 2021*; *Desgranges et al., 2022*; *Lalaouna et al., 2019*). Moreover, several 5′UTR-derived and ORF-internal RNAs have been discovered in *S. enterica* and *E. coli* (*Adams et al., 2021*; *Matera et al., 2022*). Wherever sRNAs are derived, their parental mRNAs share identical nucleotide sequences and thus can function as post-transcriptional regulators, too. Likewise, dual-function sRNAs are actually mRNAs that encode small peptides and base-pair with target RNAs *in trans* (*Raina et al., 2018*). The two events on dual-function sRNAs are mutually exclusive depending on translation efficiency, RNA stability, and temperature (*Aoyama et al., 2021*; *Raina et al., 2022*; *Balasubramanian and Vanderpool, 2013*; *Aoyama et al., 2022*). Moreover, post-transcriptional regulation via mRNA-mRNA interactions is not rare in Gram-positive bacteria (*Liu et al., 2015*; *Chen et al., 2015*; *Ignatov et al., 2020*; *Mediati et al., 2022*).

Nonetheless, whether the processing is required for the mRNA-derived sRNAs remained obscure. This study first demonstrated that *E. coli* GlnZ functions independently of its parental mRNA but acts in the same biological pathway as the translated protein. This is not simply attributable to sRNA maturation through RNase E cleavage as in the case of ArcZ (*Chao et al., 2017*) since the GlnZ precursor but not the *glnA* mRNA was capable of target regulation (*Figure 6D*). We suggest that the translation of *glnA* mRNA interferes with the base-pairing interaction because the seed region is close to the stop codon in *E. coli* K-12 (*Figure 6B*).

That said, this study does not exclude the possibility that a parental mRNA itself also interacts with its target mRNA in Gram-negative bacteria. The spacer length between the stop codon and the seed region of a 3′UTR-derived sRNA varies among species and depends on genomic loci. For example, the highly conserved CpxQ sRNA contains two seed regions, R1 and R2, downstream of the *cpxP* stop codon (*Chao and Vogel, 2016*), the former of which overlaps with its identified RNase E cleavage site. In contrast, the NarS sRNA is processed at a conserved RNase E cleavage site far upstream of the *narK* stop codon, but *E. coli* NarS is much larger than *Salmonella* NarS due to a 170 nt insertion sequence downstream of the *narK* stop codon (*Wang et al., 2020*). It needs to be explored further how the prokaryotic gene expression system coordinates translation, processing, and base-pairing events occurring on a single mRNA molecule.

## Multilayered regulation of the nitrogen assimilation pathway

The regulation of GS has been intensively studied in model enterobacteria particularly on the transcriptional regulation and the post-translational modification (*van Heeswijk et al., 2013*). As the relative levels of *glnA* mRNA and its protein product are discordant in the *E. coli hfq* mutant (*Figure 5A*), we suggest that GS is also subject to Hfq-mediated post-transcriptional regulation. However, it seems counterintuitive that *S. enterica* and *E. coli* express strikingly high levels of GS in the presence of its metabolic product (*Figure 1B*), even though the activity of GS is tightly controlled through

post-translational modification (*Huergo et al., 2013*). This raises a question of how enterobacteria coordinate the activity and expression of GS in response to nitrogen availability.

Cyanobacteria and archaea utilize different types of GS without covalent modification and have evolved unique mechanisms to regulate GS through both post-transcriptional regulation and protein-protein interactions (*Prasse and Schmitz, 2018*; *Moeller et al., 2021*; *Bolay et al., 2018*). In meta-nogenic archaea *Methanosarcina mazei*, the sRNA$_{154}$ controls the expression of GS and P$_{II}$-like signal transduction protein (*Prasse et al., 2017*). In addition, a small protein sP26 is induced upon nitrogen starvation and stimulates the activity of GS (*Gutt et al., 2021*). In *Synechocystis*, the global nitrogen regulator NtcA upregulates *glnA* in response to increased 2-OG levels and downregulates *gifA* and *gifB* encoding the GS inactivating factors IF7 and IF17, respectively (*García-Domínguez et al., 2000*). NtcA also activates the expression of NsiR4 sRNA, which represses the *gifA* gene at the post-transcriptional level (*Klähn et al., 2015*). The expression of *gifB* is enhanced by a Gln-binding ribo-switch in its 5′UTR, enabling a rapid negative feedback of GS activity in response to its product (*Klähn et al., 2018*).

Besides the Gln and Glu synthetic pathways, the branch point in the TCA cycle is a plausible target to fine-tune the metabolic balance between carbon and nitrogen as we show that GlnZ primarily regu-lates the expression of OGDH in enterobacteria. Other bacteria are known to modulate the activity of TCA cycle enzymes in response to nitrogen availability through direct protein interactions. *Cory-nebacterium glutamicum* produces high levels of Glu by inhibiting the OGDH activity with a 15 kDa protein OdhI depending on its phosphorylation state (*Niebisch et al., 2006*). In *V. cholerae*, the VcdRP dual-functional sRNA represses the PTS-coding mRNAs while the 29-aa small protein VcdP activates citrate synthase GltA to accumulate 2-OG and Glu (*Venkat et al., 2021*).

In summary, this study adds another layer to the complex regulatory system of nitrogen assimilation in enterobacteria; the *glnA* mRNA expresses both GS and the post-transcriptional regulator GlnZ to fulfill the supply and demand of the nitrogen reservoir. Still, even with the help of the transcriptional regulator Nac in *E. coli*, GlnZ does not fully account for the strong repression of SucA expression upon nitrogen limitation. As many bacteria and archaea employ various regulatory RNAs and proteins, future studies will identify more factors for the redundant control of the central metabolic pathways in the model microorganisms.

# Materials and methods

## Bacterial strains

*S. enterica* serovar Typhimurium strain SL1344 (JVS-1574) and *E. coli* strain BW25113 were used as WT strains. The strains used in this study are listed in *Supplementary file 2*. Bacterial cells were grown at 37°C with reciprocal shaking at 180 rpm in LB Miller medium (BD Biosciences) or MOPS minimal medium (*Neidhardt et al., 1974*). As a carbon source, MOPS minimal medium was supple-mented with 0.2% glucose or 20 mM sodium pyruvate. Where appropriate, the nitrogen source of the MOPS minimal medium was substituted by 0.01% of ammonium chloride or 5 mM Gln (prepared freshly to avoid spontaneous hydrolysis) for 0.1% ammonium chloride. For SL1344 and its derivatives, 40 μM histidine was added into MOPS minimal medium, which does not interfere with GS expres-sion. Optical density (OD$_{600}$) was automatically monitored throughout the growth every 10 min using OD-Monitor C&T (TAITEC). Where appropriate, media were supplemented with antibiotics at the following concentrations: 100 μg/ml ampicillin (Ap), 50 μg/ml kanamycin (Km), and 20 μg/ml chloram-phenicol (Cm).

## Plasmid construction

A list of all plasmids and oligonucleotides used in this study can be found in *Supplementary file 3 and 4*. Constitutive expression plasmids, arabinose-inducible pBAD expression plasmids, and trans-lational fusion plasmids were derived from pZE12-luc, pKP8-35, and pXG30-sf, respectively (*Urban and Vogel, 2007*; *Papenfort et al., 2006*; *Corcoran et al., 2012*). Single-nucleotide mutations were introduced by inverse PCR using KOD-Plus-Neo DNA polymerase (Toyobo) and overlapping primer pairs (*Supplementary file 4*) followed by DpnI digestion. The nucleotide sequences of the plasmid inserts can be found in *Supplementary file 5*.

## Strain construction

The *glnZ* deletion strains were constructed by the $\lambda$ Red system (***Datsenko and Wanner, 2000***). The *glnA* 3'UTR was deleted using pKD4 as a template and primer pairs, MMO-0371/0372 for *Salmonella* and MMO-0401/0402 or MMO-0401/0583 for *E. coli*, respectively. The resulting Km resistant strains were confirmed by PCR, and the mutant loci were transduced into appropriate genetic backgrounds by P22 and P1 phages in *Salmonella* and *E. coli*, respectively. The temperature-sensitive plasmid pCP20 expressing FLP recombinase was used to eliminate the resistance gene from the chromosome.

The $\Delta hfq::cat$, *ams-1*::Tn*10*, and *rnc14*::Tn*10* loci were transduced from *E. coli* strains TM587 (***Morita et al., 2005***), TM151 (***Morita et al., 2006***), and ST201 (***Sunohara et al., 2004***) into appropriate genetic backgrounds by P1 phage, respectively.

The *glnZ* locus in BW25113 was substituted for *glnZ* from O157 or O111 by scar-less mutagenesis using a two-step $\lambda$ Red recombination system (***Blank et al., 2011***). The DNA fragment containing a CmR resistance marker and a I-SceI recognition site was amplified with primer pairs MMO-0401/0451 using pWRG100 plasmid as a template and was integrated into the BW25113 chromosome by $\lambda$ Red recombinase expressed from pKD46. The resultant CmR mutant was transformed by pWRG99, and the heterogeneous *glnZ* allele from O157 or O111 was amplified with MMO-0786/0787 and integrated into the chromosome by $\lambda$ Red recombinase expressed from pWRG99. The resultant recombinants were selected on LB agar plate supplemented with Ap and 2 μg/ml of anhydrotetracycline to express I-SceI endonuclease. The successful recombinants were confirmed by Cm sensitivity, PCR, and sequencing.

The 3xFLAG epitope tag at the C-terminus of *ntrC* was amplified with primer pairs MMO-0757/0758 using pSUB11 (***Uzzau et al., 2001***) as a template and was introduced into the chromosome by the $\lambda$ Red system. The resulting Km resistant strains were confirmed by PCR, and the mutant locus was transduced into appropriate genetic backgrounds by P1 phage.

## GFP fluorescence quantification

*E. coli* Δ*glnZ* transformants harboring a combination of the translational fusions and the GlnZ constitutive expression plasmids (***Supplementary file 3, 5 and 7***) were inoculated from single colonies in 500 μl LB medium containing Ap and Cm in 96-well deep well plates (Thermo Fisher Scientific) and were grown overnight at 37°C with rotary shaking at 1200 rpm in DWMax M-BR-032P plate shaker (Taitec). The 100 μl overnight cultures were dispensed into 96-well optical bottom black microtiter plates (Thermo Fisher Scientific). $OD_{600}$ and fluorescence (excitation at 485 nm and emission at 535 nm with dichroic mirror of 510 nm, fixed gain value of 50) were measured using Spark plate reader (Tecan). The relative fluorescence units were calculated by subtracting the autofluorescence of bacterial cells without *gfp* grown in the same condition and normalized by $OD_{600}$.

## Northern blot

Total RNA was isolated using the TRIzol reagent (Invitrogen), treated with TURBO DNase (Invitrogen), and precipitated with cold ethanol. RNA was quantified using NanoDrop One (Invitrogen). Total RNA (5 μg) was separated by gel electrophoresis on 6% or 8% polyacrylamide/7 M urea gels in 1×TBE buffer for 2.5 hr at 300 V using Biometra Eco-Maxi system (Analytik-Jena). DynaMarker RNA Low II ssRNA fragment (BioDynamics Laboratory) was used as a size marker. RNA was transferred from the gel onto Hybond N+membrane (Cytiva) by electroblotting for 1 hr at 50 V using the same device. The membrane was crosslinked with transferred RNA by 120 mJ/cm$^2$ UV light, incubated for prehybridization in Rapid-Hyb buffer (Cytiva) at 42°C for 1 hr, and then incubated for hybridization with a [$^{32}$P]-labeled probe (***Supplementary file 4***) at 42°C overnight. The membrane was washed in three 15 min steps in 5×SSC/0.1% SDS, 1×SSC/0.1% SDS, and 0.5×SSC/0.1% SDS buffers at 42°C. Signals were visualized on Typhoon FLA7000 scanner (GE Healthcare) and quantified using Image Quant TL software (GE Healthcare).

For northern blot analysis of mRNA, total RNA samples (2 μg) were resolved by 1.5% agarose gel electrophoresis in the presence of formaldehyde and blotted onto a Nylon Membrane, positively charged (Roche). DynaMarker Prestain Marker for RNA high (BioDynamics Laboratory) was used as a size marker. The RNAs were visualized by using a detection system with digoxigenin (DIG) (Roche) and then captured using the imaging system ChemiDoc XRS Plus (Bio-Rad). The antisense RNA probes corresponding to the 3'-end portion of *glnA* CDS were prepared by the DIG RNA Labeling Kit (Roche).

## Western blot

Bacteria culture was collected by centrifugation for 5 min at 5000 × $g$ at 4°C, and the pellet was dissolved in 1× protein loading buffer to a final concentration of 0.01 OD/μl. After heating for 5 min at 95°C, 0.002 OD of whole-cell samples were separated on 7.5% TGX gels (Bio-Rad). Proteins were transferred onto Hybond P PVDF 0.2 membranes (Cytiva), and membranes were blocked for 10 min in Bullet Blocking One (Nacalai Tesque, Kyoto, Japan) and were incubated for 1 hr at RT or overnight at 4°C with rabbit polyclonal α-SucA (1:10,000, Tanpaku Seisei Co., Gunma, Japan), α-SucB (1:10,000, Tanpaku Seisei Co., Gunma, Japan), α-AceE (1:10,000, Tanpaku Seisei Co., Gunma, Japan), α-GlnA (1:5000, raised against a synthetic peptide CAHQVNAEFFEEGKMFDGSS, Sigma-Aldrich), α-GroEL (1:10,000, Sigma-Aldrich #G6532), and mouse monoclonal α-FLAG (1:5000, Sigma-Aldrich #F1804) diluted in Bullet Blocking One. Membranes were washed three times for 15 min in 1×TBST buffer at RT. Then, membranes were incubated for 1 hr at RT with secondary anti-mouse or anti-rabbit HRP-linked antibodies (Cell Signaling Technology #7076 or #7074; 1:5000) diluted in Bullet Blocking One and were washed three times for 15 min in 1×TBST buffer. Chemiluminescent signals were developed using Amersham ECL Prime reagents (Cytiva), visualized on LAS4000 (GE Healthcare), and quantified using Image Quant TL software.

## RNA-seq analysis

Biological duplicates of *S.* Typhimurium SL1344 strain harboring pKP-8-35 or pBAD-GlnZ1 plasmids were grown to exponential phase ($OD_{600}$ ~1.0) in LB medium. sRNA expression was induced by adding 0.2% L-arabinose for 10 min. Cell cultures were immediately mixed with 20% (v/v) of stop solution (95% ethanol, 5% phenol) and snap-frozen in liquid nitrogen. Total RNA was isolated using the TRIzol reagent (Invitrogen) and treated with TURBO DNase (Invitrogen). Ribosomal RNA was depleted using NEBNext rRNA Depletion Kit (Bacteria), and RNA integrity was confirmed using TapeStation (Agilent). Directional cDNA libraries were prepared using the NEBNext Ultra II Directional RNA Library Prep Kit for Illumina (#7760L). The libraries were sequenced using NovaSeq 6000 platform in the single-read mode for 101 cycles.

Reads were mapped to the *S.* Typhimurium SL1344 reference genome (GenBank accession number FQ312003.1) using bowtie2 version 2.3.2, and HTSeq version 0.5.3p3 was used to generate the count matrix. Reads mapping to annotated coding sequences were counted, normalized (CPM), and transformed ($log_2$ fold change). Differential expression between the conditions was tested using edgeR version 3.28.0. Genes with a $log_2$ fold change >2.0 and FDR adjusted p-value <0.01 were defined as differentially expressed. The RNA-seq data have been deposited in DDBJ DRA under accession number DRA012682.

## Acknowledgements

The authors would like to thank Jörg Vogel for critical reading, Maxence Lejars, Takeshi Kanda, and Nozomu Obana for discussion and help with preparing figures, Natsuko Shirai for technical assistance, and NBRP-*E. coli* at NIG for providing *E. coli* strains. This work was partly performed in the Cooperative Research Project Program of the Medical Institute of Bioregulation, Kyushu University. FUNDING: This study was supported by JSPS KAKENHI grant numbers JP16H06279 (PAGS), JP19H03464, JP19KK0406, JP21K19063 to MM, and JP22H02236 to TK. Research in the Miyakoshi laboratory is funded by The Waksman Foundation of Japan and Takeda Science Foundation. MM is supported by Tomizawa Jun-ichi & Keiko Fund of Molecular Biology Society of Japan for Young Scientist.

## Additional information

### Funding

| Funder | Grant reference number | Author |
| --- | --- | --- |
| Japan Society for the Promotion of Science | JP19H03464 | Masatoshi Miyakoshi |

| Funder | Grant reference number | Author |
|---|---|---|
| Japan Society for the Promotion of Science | JP19KK0406 | Masatoshi Miyakoshi |
| Japan Society for the Promotion of Science | JP21K19063 | Masatoshi Miyakoshi |
| Japan Society for the Promotion of Science | JP22H02236 | Kan Tanaka |
| Japan Society for the Promotion of Science | JP16H06279 | Hiroki Takahashi Tetsuya Hayashi |
| Waksman Foundation of Japan | | Masatoshi Miyakoshi |
| Takeda Medical Research Foundation | | Masatoshi Miyakoshi |

The funders had no role in study design, data collection and interpretation, or the decision to submit the work for publication.

## Author contributions
Masatoshi Miyakoshi, Conceptualization, Resources, Data curation, Formal analysis, Supervision, Funding acquisition, Validation, Investigation, Visualization, Methodology, Writing - original draft, Project administration, Writing – review and editing; Teppei Morita, Validation, Investigation, Visualization, Writing – review and editing; Asaki Kobayashi, Anna Berger, Investigation; Hiroki Takahashi, Data curation, Formal analysis, Funding acquisition; Yasuhiro Gotoh, Data curation, Formal analysis; Tetsuya Hayashi, Formal analysis, Funding acquisition, Writing – review and editing; Kan Tanaka, Resources, Funding acquisition, Writing – review and editing

## Author ORCIDs
Masatoshi Miyakoshi (ID) http://orcid.org/0000-0002-4901-2809
Teppei Morita (ID) http://orcid.org/0000-0001-5057-5687
Hiroki Takahashi (ID) http://orcid.org/0000-0001-5627-1035
Tetsuya Hayashi (ID) http://orcid.org/0000-0001-6366-7177
Kan Tanaka (ID) http://orcid.org/0000-0001-7560-7884

## Decision letter and Author response
Decision letter https://doi.org/10.7554/eLife.82411.sa1
Author response https://doi.org/10.7554/eLife.82411.sa2

# Additional files

## Supplementary files
• Supplementary file 1. Genes downregulated upon GlnZ1 overexpression in *Salmonella* Typhimurium SL1344.
• Supplementary file 2. Bacterial strains used in this study.
• Supplementary file 3. Plasmids used in this study.
• Supplementary file 4. DNA oligonucleotides used in this study.
• Supplementary file 5. Inserts of GlnZ mutant plasmids.
• Supplementary file 6. Details of GFP fusion plasmids.
• Supplementary file 7. Inserts of GFP fusion plasmids.
• MDAR checklist

## Data availability
The RNA-seq data have been deposited in DDBJ SRA under accession number DRA012682.

The following dataset was generated:

| Author(s) | Year | Dataset title | Dataset URL | Database and Identifier |
|-----------|------|---------------|-------------|-------------------------|
| Miyakoshi M | 2021 | Regulatory network analysis of mRNA 3'UTRs in Salmonella | https://ddbj.nig.ac.jp/resource/sra-submission/DRA012682 | DDBJ Sequence Read Archive, DRA012682 |

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
