## [Editor Report]

This important study supports the role of a regulatory RNA to adjust to the changing availability of nitrogen by controlling expression of key enzymes of the nitrogen assimilation pathway in *Enterobacteriaceae*. The evidence supporting the conclusions is convincing, with state-of-the-art genetic and biochemical assays. The work will be of interest to scientists within the field of microbial gene regulation.

---

## [Decision Letter]

**Decision letter after peer review:**

Thank you for submitting your article "Glutamine synthetase mRNA releases a small RNA from its 3´UTR to regulate carbon/nitrogen metabolic balance" for consideration by *eLife*. Your article has been reviewed by 3 peer reviewers, including Alexander J. Westermann as Reviewing Editor and Reviewer #1, and the evaluation has been overseen by Satyajit Rath as the Senior Editor. The following individual involved in review of your submission has agreed to reveal their identity: Stephan Klaehn (Reviewer #2).

Essential revisions:

1. There remain some open questions as to the mechanism of GlnZ-mediated target control: For example, GlnZ binds its targets sucA and aceE mRNAs far upstream of the ribosome binding site. The authors should expand on this interesting observation: might this sequence stretch be a stabilizing element or does it harbor a potential translational enhancer? At the least, this should be discussed in the main text. Can the authors also please comment on whether or not GlnZ-mediated repression of sucA extends to the downstream genes in that operon?

2. The role of Hfq in GlnZ-mediated target control remains unclear and needs to be experimentally assessed. The 3'UTR of glnA (=GlnZ) is known to associate with Hfq (PMIDs: 15718303; 19333007) and existing RIL-seq data (PMID: 27588604) further suggest the existence of trimeric complexes (GlnZ:Hfq:target) in the bacterial cell. However, the experiment presented in Figure 5A does not support the authors' conclusion that Hfq influences processing but not stability of GlnZ since it 1) only records steady-state levels and 2) is based on strong overexpression of the sRNA (here: from the pBAD promoter), which may lead to misinterpretation of Hfq-dependence. Probing GlnZ expression in an hfq deletion mutant in response to nitrogen starvation would be more significant to draw a conclusion. Either way, it remains also unclear whether Hfq is involved in GlnZ-mediated target regulation. Could the authors test if the (mature) GlnZ sRNA is still capable of repressing e.g. sucA in an hfq deletion background?

3. As to the metabolic consequences, the authors should either include data to show any downstream effect of GlnZ-mediated sucA regulation (e.g. 2-OG levels, metabolic flux through OGDH, etc.) or at least tone down the statement that GlnZ redirects the carbon flow from the TCA cycle to the nitrogen assimilation pathway.

Suggested improvements:

In general, we strongly recommend that the authors set their work more apart from that by Walling et al. (meanwhile published in NAR) by focusing on the unique aspects of their study. This would best be achieved by putting more weight on the comparison of GlnZ between *Salmonella* and *E. coli*. For example, the authors could experimentally validate *Salmonella*-specific targets of GlnZ to highlight species-specific differences of this sRNA's regulon. Moreover, while the 2 GlnZ isoforms of *Salmonella* acted largely redundant in the presented assays, would they have unique roles in the regulation of some of the *Salmonella*-specific targets? Differences in GlnZ-mediated regulation across enterobacteria could further be complemented by computational efforts, e.g. is the presence of an additional promoter within the glnA 3' end reflected in the phylogenetic classification of the different homologs?

The assumed role of translation in interfering with the basepairing to target mRNAs is interesting, as it could be showcase for additional 3' UTR-derived sRNAs, and should therefore be addressed in greater detail. The authors could test it experimentally, e.g. by mutating the start codon of glnA (in the mut3 background, so RNase E cleavage remains abrogated) to see if such a construct would be able to repress SucA:GFP to the same levels as does the GlnZ sRNA. Alternatively, given the sequence variation in different bacterial species, would an increased distance from the translational stop codon (and thus, from the translation machinery) result in enhanced target control by glnA?

For easier reading, we recommend to swap sentences 2 and 3 of the abstract.

L. 41: either add an article or remove "pathway".

L. 47: "a" should be replaced by "the".

L. 47-49: The sentence "For a facultative intracellular pathogen…" appears somewhat misplaced here because genes encoding GS are first introduced in the following paragraph. Please move it further down.

L. 61: "is perceived as nitrogen limitation…"

L. 70: the concept of small RNAs should be introduced in more detail.

L.73: tone down the statement "In general, alternative σ factors drive the transcription of small RNAs". This sentence implicates that it is the major task of alternative σ factors and that they might actually be required for the transcription of sRNAs. Please rephrase.

L. 77-78 and l. 81-82: "The only RpoN-dependent sRNA known so far in enterobacteria is GlmY" and "there may be more than one RpoN-dependent sRNA" – In the present introduction the authors aim to introduce their work on GlnZ in a frame with its RpoN-dependence. However, it should be sufficient for readers to introduce post-transcriptional regulation by sRNAs and distinguish trans-encoded ones from those which are derived from mRNA processing. The fact that GlnZ requires RpoN can of course be mentioned, but should not be used as a motivation for this study.

L. 83: "mRNA 3´UTRs" please rephrase to "3'UTRs of mRNAs".

L. 83, 84: replace "an emerging class of post-transcriptional regulators".

L. 88: "a sponge" instead of "the sponge".

L. 90: "gltA mRNA".

L. 92: introduce Hfq.

L. 120: "To clarify whether glnA mRNA and GlnZ sRNA are also induced upon nitrogen limitation" – please rephrase to "To clarify whether the level of glnA mRNA and GlnZ sRNA correlate with protein accumulation".

L. 176: “sucA is located…" – please rephrase to “The sucA gene is located”…".

L. 244-245: "we ectopically expressed the glnA mRNA of *S. enterica* and *E. coli* from an arabinose-inducible pBAD promoter" – please rephrase to "we ectopically expressed the glnA gene of *S. enterica* and *E. coli* from a pBAD plasmid, i.e. driven by an arabinose-inducible promoter".

Line 262: change "permissible" to "permissive".

L. 267: it is not required to consecutively write "GlnZ sRNA". GlnZ would be sufficient here (and also in other parts of the manuscript).

L. 302ff: this paragraph needs some logical reconstruction/rephrasing. In particular, the classification of unique regulatory mechanisms for nitrogen metabolism and glutamine synthetase should be addressed. We suggest to mention that the bulk of our knowledge on the NtrBC system and post-translational modifications of GS has been obtained from model enterobacteria. Apart from this, exceptions have been identified in the archaeal domain, but also among several bacterial phyla such as Firmicutes and Cyanobacteria. Further, we suggest to clearly distinguish transcriptional regulation from post-transcriptional mechanisms, but also to mention biochemical mechanisms that are unique to a specific group of microorganisms as well (IFs in cyanobacteria, sP26 in Archaea). Currently, the paragraph condenses all these different facets in only a few sentences.

L. 310: Please rephrase "the global nitrogen cycle" – what the authors mean here is nitrogen assimilation/metabolism. With "global nitrogen cycle" typically the biogeochemical cycle consisting of N2 fixation, nitrification, anamox etc. is meant.

L. 367-375: In Walling et al., not only do they show that GlnZ itself is a substrate of RNase III (as the present authors rightfully point out in line 366), but GlnZ seems to also direct RNase III cleavage of sucA and aceF mRNAs. This could also be mentioned and cited in this section of the discussion, when the present authors describe their own findings on RNase III and SucA and AceF.

Ref. "Hör et al., 2020" (p. 26) is mistakenly cited as a preprint.

Figure 1B-C: Instead of labeling the blots with 1-4 we suggest to use the respective condition names. This would increase comprehensibility and it has also been done in all other figures.

Figure 1D: How is full-length GlnZ ('GlnZ1') defined? Another band at ~200 nt is also very abundant.

Figure 1, legend: Please mention SucA and SucB in brackets when you refer to "OGDH subunits" as the acronym OGDH is not shown in the figure. In addition to GlnA and GltBD please also add the abbreviations used in the text until this point (e.g. GS, GOGAT) to the figure legend.

Figure 3A: For the x-axis, we suggest to use the term "pJV300" or "control" instead of "vector".

Figure 4B: As the effects are rather subtle, the authors should provide robust quantification of the western blot data (including replicates).

Figure 6C: "Fold change" (on the y-axis) should be relabeled as "Fold repression" to avoid confusion (positive FC actually meaning downregulation).

---

## [Author Response]

Essential revisions:1. There remain some open questions as to the mechanism of GlnZ-mediated target control: For example, GlnZ binds its targets sucA and aceE mRNAs far upstream of the ribosome binding site. The authors should expand on this interesting observation: might this sequence stretch be a stabilizing element or does it harbor a potential translational enhancer? At the least, this should be discussed in the main text. Can the authors also please comment on whether or not GlnZ-mediated repression of sucA extends to the downstream genes in that operon?

The GlnZ-mediated regulation of *sucA* resembles the case of *sdhC* repression by distantly binding sRNA Spot42, which recruits Hfq to the translation initiation region (Desnoyers and Masse, 2012). We found a purine-rich sequence upstream of *sucA* SD sequence, which potentially binds to Hfq. We have clarified that this sequence is crucial for GlnZ-mediated repression. We have added new Figure 3 —figure supplement 1 and a relevant explanation to the revised manuscript (Lines 271-285).

In addition, we ectopically expressed GlnZ in *E. coli glnZ* mutant to analyze the expression levels of chromosomally expressed targets. We observed striking repression of SucA and AceE upon GlnZ overexpression, but the level of SucB protein was not significantly affected, showing that GlnZ-mediated regulation of *sucA* does not extend to the downstream genes. We have added new Figure 5B and a relevant explanation to the revised manuscript (Lines 258-270).

2. The role of Hfq in GlnZ-mediated target control remains unclear and needs to be experimentally assessed. The 3'UTR of glnA (=GlnZ) is known to associate with Hfq (PMIDs: 15718303; 19333007) and existing RIL-seq data (PMID: 27588604) further suggest the existence of trimeric complexes (GlnZ:Hfq:target) in the bacterial cell. However, the experiment presented in Figure 5A does not support the authors' conclusion that Hfq influences processing but not stability of GlnZ since it 1) only records steady-state levels and 2) is based on strong overexpression of the sRNA (here: from the pBAD promoter), which may lead to misinterpretation of Hfq-dependence. Probing GlnZ expression in an hfq deletion mutant in response to nitrogen starvation would be more significant to draw a conclusion. Either way, it remains also unclear whether Hfq is involved in GlnZ-mediated target regulation. Could the authors test if the (mature) GlnZ sRNA is still capable of repressing e.g. sucA in an hfq deletion background?

We analyzed the GlnZ expression in *E. coli hfq* deletion mutant grown in minimal media as in Figure 1. We observed the expression of both GlnA protein and *glnA* mRNA but only faint signals of GlnZ, clarifying that Hfq is required for GlnZ biogenesis. Intriguingly, we also found that the expression of GlnA is rather elevated in the *hfq* mutant, implying the post-transcriptional regulation of *glnA*. We have added new Figure 5A and a relevent explanation (Lines 250-257) and discussion (Lines 447-453) to the revised manuscript.

We also tested whether the ectopic expression of GlnZ repressed the expression of SucA and AceE proteins in the *E. coli* ∆*glnZ* ∆*hfq* double mutant. Even though GlnZ was expressed at lower levels, GlnZ did not affect the chromosomally expressed targets in the absence of Hfq (Figure 5B). This result shows that Hfq is involved in the stability of GlnZ and the target regulation by GlnZ. However, when the *E. coli glnA* mRNA was ectopically expressed from the arabinose-inducible promoter, no GlnZ was observed in the absence of Hfq (Figure 5C). This result shows that Hfq is essential for the processing of the *glnA* mRNA. The previous version of Figure 5A has been replaced with Figure 5C in the revised manuscript, and we have added a relevant to the revised manuscript (Lines 258-270).

3. As to the metabolic consequences, the authors should either include data to show any downstream effect of GlnZ-mediated sucA regulation (e.g. 2-OG levels, metabolic flux through OGDH, etc.) or at least tone down the statement that GlnZ redirects the carbon flow from the TCA cycle to the nitrogen assimilation pathway.

We have toned down the statement about the metabolic consequences throughout the revised manuscript.

Suggested improvements:In general, we strongly recommend that the authors set their work more apart from that by Walling et al. (meanwhile published in NAR) by focusing on the unique aspects of their study. This would best be achieved by putting more weight on the comparison of GlnZ between Salmonella and *E. coli*.

We have focused on the changes in gene expression of central carbon metabolism in response to nitrogen limitation. In addition to the post-transcriptional regulator GlnZ, we also show that the transcriptional regulator Nac is involved in the repression of *suc* operon under nitrogen limiting conditions in *E. coli*. Moreover, we have clarified the involvement of RNase E in the post-transcriptional regulation and show that RNase E is critical for the processing of GlnZ from the *glnA* mRNA, and the *glnA* mRNA devoid of processing is unable to regulate the target mRNA. To our knowledge, this is the first study that showed a mutation in RNase E cleavage site abrogated the regulatory function of 3´UTR-derived sRNAs without affecting the seed region.

To highlight the *Salmonella*-specific targets, we have integrated previous Supplementary Figure 1 into Figure 3C and D in the revised manuscript. *Salmonella* GlnZ interacts with the *deoD* gene within the first three codons, but *E. coli deoD* contains synonymous mutations. We also show the post-transcriptional regulation of *aceE* by reporter analysis (Figure 3E and F) and the expression changes of the AceE protein western blot, which has now been shown in Figure 5B of the revised manuscript. We identified the single nucleotide substitution in GlnZ from *Salmonella* and *E. fergusonii* which causes the dysregulation of *aceE* (Figure 3F and Figure 3 —figure supplement 2), as discussed in Lines 373-383.

For example, the authors could experimentally validate Salmonella-specific targets of GlnZ to highlight species-specific differences of this sRNA's regulon. Moreover, while the 2 GlnZ isoforms of Salmonella acted largely redundant in the presented assays, would they have unique roles in the regulation of some of the Salmonella-specific targets? Differences in GlnZ-mediated regulation across enterobacteria could further be complemented by computational efforts, e.g. is the presence of an additional promoter within the glnA 3' end reflected in the phylogenetic classification of the different homologs?

Using the CopraRNA program, we have searched for additional genes which are specifically targeted by the longer *Salmonella* GlnZ isoform. However, the translational fusions of predicted candidates showed no expression changes upon ectopic expression of *Salmonella* GlnZ.

The assumed role of translation in interfering with the basepairing to target mRNAs is interesting, as it could be showcase for additional 3' UTR-derived sRNAs, and should therefore be addressed in greater detail. The authors could test it experimentally, e.g. by mutating the start codon of glnA (in the mut3 background, so RNase E cleavage remains abrogated) to see if such a construct would be able to repress SucA:GFP to the same levels as does the GlnZ sRNA. Alternatively, given the sequence variation in different bacterial species, would an increased distance from the translational stop codon (and thus, from the translation machinery) result in enhanced target control by glnA?

Thank you so much for your suggestion. We are analyzing whether some other 3' UTR-derived sRNAs keep their functions when transcribed from their parental mRNAs as discussed in Lines 431-441. In a follow-up study, we will clarify how translation and processing events are coordinated around the stop codon.

For easier reading, we recommend to swap sentences 2 and 3 of the abstract.

The abstract has been rewritten in the revised manuscript.

L. 41: either add an article or remove "pathway".

Corrected.

L. 47: "a" should be replaced by "the".

Corrected.

L. 47-49: The sentence "For a facultative intracellular pathogen…" appears somewhat misplaced here because genes encoding GS are first introduced in the following paragraph. Please move it further down.

The second paragraph has been rewritten in the revised manuscript.

L. 61: "is perceived as nitrogen limitation…"

Corrected.

L. 70: the concept of small RNAs should be introduced in more detail.

The introduction of small RNAs has been added to the revised manuscript (Lines 81-84).

L.73: tone down the statement "In general, alternative σ factors drive the transcription of small RNAs". This sentence implicates that it is the major task of alternative σ factors and that they might actually be required for the transcription of sRNAs. Please rephrase.

Thank you very much for pointing out our mistake. We have rewritten the sentence in the revised manuscript (Lines 79-81).

L. 77-78 and l. 81-82: "The only RpoN-dependent sRNA known so far in enterobacteria is GlmY" and "there may be more than one RpoN-dependent sRNA" – In the present introduction the authors aim to introduce their work on GlnZ in a frame with its RpoN-dependence. However, it should be sufficient for readers to introduce post-transcriptional regulation by sRNAs and distinguish trans-encoded ones from those which are derived from mRNA processing. The fact that GlnZ requires RpoN can of course be mentioned, but should not be used as a motivation for this study.

We agree with this comment. We have deleted these sentences in the revised manuscript.

L. 83: "mRNA 3´UTRs" please rephrase to "3'UTRs of mRNAs".

Corrected.

L. 83, 84: replace "an emerging class of post-transcriptional regulators".

Corrected.

L. 88: "a sponge" instead of "the sponge".

Corrected.

L. 90: "gltA mRNA".

Corrected.

L. 92: introduce Hfq.

The introduction of Hfq has been added to the revised manuscript (Lines 81-84).

L. 120: "To clarify whether glnA mRNA and GlnZ sRNA are also induced upon nitrogen limitation" – please rephrase to "To clarify whether the level of glnA mRNA and GlnZ sRNA correlate with protein accumulation".

Corrected.

L. 176: “sucA is located…" – please rephrase to “The sucA gene is located”…".

Corrected.

L. 244-245: "we ectopically expressed the glnA mRNA of S. enterica and *E. coli* from an arabinose-inducible pBAD promoter" – please rephrase to "we ectopically expressed the glnA gene of S. enterica and *E. coli* from a pBAD plasmid, i.e. driven by an arabinose-inducible promoter".

Corrected.

Line 262: change "permissible" to "permissive".

Corrected.

L. 267: it is not required to consecutively write "GlnZ sRNA". GlnZ would be sufficient here (and also in other parts of the manuscript).

Corrected.

L. 302ff: this paragraph needs some logical reconstruction/rephrasing. In particular, the classification of unique regulatory mechanisms for nitrogen metabolism and glutamine synthetase should be addressed. We suggest to mention that the bulk of our knowledge on the NtrBC system and post-translational modifications of GS has been obtained from model enterobacteria. Apart from this, exceptions have been identified in the archaeal domain, but also among several bacterial phyla such as Firmicutes and Cyanobacteria. Further, we suggest to clearly distinguish transcriptional regulation from post-transcriptional mechanisms, but also to mention biochemical mechanisms that are unique to a specific group of microorganisms as well (IFs in cyanobacteria, sP26 in Archaea). Currently, the paragraph condenses all these different facets in only a few sentences.

We have largely written this paragraph in the revised manuscript (Lines 444-481).

L. 310: Please rephrase "the global nitrogen cycle" – what the authors mean here is nitrogen assimilation/metabolism. With "global nitrogen cycle" typically the biogeochemical cycle consisting of N2 fixation, nitrification, anamox etc. is meant.

Corrected.

L. 367-375: In Walling et al., not only do they show that GlnZ itself is a substrate of RNase III (as the present authors rightfully point out in line 366), but GlnZ seems to also direct RNase III cleavage of sucA and aceF mRNAs. This could also be mentioned and cited in this section of the discussion, when the present authors describe their own findings on RNase III and SucA and AceF.

We performed an additional experiment on the role of RNase III in the processing of GlnZ (now shown in Figure 6 —figure supplement 1.) and have written this paragraph in the revised manuscript (Lines 400-408).

Ref. "Hör et al., 2020" (p. 26) is mistakenly cited as a preprint.

Corrected.

Figure 1B-C: Instead of labeling the blots with 1-4 we suggest to use the respective condition names. This would increase comprehensibility and it has also been done in all other figures.

Corrected.

Figure 1D: How is full-length GlnZ ('GlnZ1') defined? Another band at ~200 nt is also very abundant.

We denoted GlnZ as the abundant transcript whose size matches the length of 3´UTR of *Salmonella glnA* as explained in Lines 125-129. The ~200 nt transcript represents a precursor of GlnZ1, which is generated by the cleavage by RNase E at the site upstream of *glnA* stop codon. We have added a relevant explanation to the revised manuscript (Lines 322-324).

Figure 1, legend: Please mention SucA and SucB in brackets when you refer to "OGDH subunits" as the acronym OGDH is not shown in the figure. In addition to GlnA and GltBD please also add the abbreviations used in the text until this point (e.g. GS, GOGAT) to the figure legend.

Corrected.

Figure 3A: For the x-axis, we suggest to use the term "pJV300" or "control" instead of "vector".

Corrected.

Figure 4B: As the effects are rather subtle, the authors should provide robust quantification of the western blot data (including replicates).

Corrected.

Figure 6C: "Fold change" (on the y-axis) should be relabeled as "Fold repression" to avoid confusion (positive FC actually meaning downregulation).

Corrected.